# Towards Generalizable Context-aware Anomaly Detection: A Large-scale Benchmark in Cloud Environments

## Abstract

Anomaly detection in cloud environments remains both critical and challenging. Existing context-level benchmarks typically focus on either metrics or logs and often lack reliable annotation, while most detection methods emphasize point anomalies within a single modality, overlooking contextual signals and limiting real-world applicability. Constructing a benchmark for context anomalies that combines metrics and logs is inherently difficult: reproducing anomalous scenarios on real servers is often infeasible or potentially harmful, while generating synthetic data introduces the additional challenge of maintaining cross-modal consistency. We introduce CloudAnoBench, a large-scale benchmark for context anomalies in cloud environments, comprising 28 anomalous scenarios and 16 deceptive normal scenarios, with 1,252 labeled cases and roughly 200,000 log and metric entries. Compared with prior benchmarks, CloudAnoBench exhibits higher ambiguity and greater difficulty, on which both prior machine learning methods and vanilla LLM prompting perform poorly. To demonstrate its utility, we further propose CloudAnoAgent, an LLM-based agent enhanced by symbolic verification that integrates metrics and logs. This agent system achieves substantial improvements in both anomaly detection and scenario identification on CloudAnoBench, and shows strong generalization to existing datasets. Together, CloudAnoBench and CloudAnoAgent lay the groundwork for advancing context-aware anomaly detection in cloud systems.

## 1 Introduction

Ensuring the stability and availability of large-scale cloud systems is of great importance (Kazemzadeh & Jacobsen, 2009; Bu et al., 2018; Zhang et al., 2015). Accurate detection methods that can also identify among anomaly scenarios are essential to mitigate potential losses (Zhang et al., 2018; Barbhuiya et al., 2018a). Large-scale cloud systems usually generate abundant logs and expose various metrics, both of which serve as some of the most valuable data sources for anomaly detection (Lin et al., 2016; Nandi et al., 2016).

Numerous benchmarks have been proposed for cloud anomaly detection such as (Oliner & Stearley, 2007; Xu et al., 2009; Akmeemana et al., 2025). However, most existing research and benchmarks for cloud anomaly detection have focused on point anomalies, where deviations are identified in isolation within a single modality, such as metrics or logs. Although these benchmarks have provided the community with relevant evaluation testbeds, they capture only a narrow slice of the anomaly landscape and often fail to reflect the complexity of real cloud environments. In practice, anomalies emerge from the interplay between multiple system components and are best understood in the context of both logs and metrics. Context anomalies fill this gap by modeling dependencies across modalities and incorporating richer scenario-level information. These anomalies account for more realistic cases such as failures caused by combined metric trends and log events. Existing context anomaly data sets (Makanju et al., 2009; Islam et al., 2025) focus on a singular modality that captures either metrics or logs and lacks clear annotation. This underlines the importance of novel improved contextual anomaly benchmarks that represent real-world operational demands.

Despite the practical relevance of contextual anomalies in cloud environments, constructing a benchmark that faithfully represents these anomalies is a challenging task. Many critical safety anomaly scenarios **cannot be replicated safely without irreversibly damaging the infrastructure**. Additionally, synthetic anomaly data needs to be generated carefully to **capture the nuanced relationships between metrics and logs** while being representative of real world incidences. These difficulties highlight why existing datasets have fallen short and why creating a large-scale, multimodal benchmark remains a fundamental challenge.

To address these challenges, we introduce CLOUDANOBENCH[1], a large-scale benchmark for context-aware anomaly detection in cloud environments. Unlike existing benchmarks, it jointly incorporates both metrics and logs, comprising 1,252 scenario labeled cases across 28 anomalous scenarios and 16 deceptive normal scenarios (approximately 200K lines), with explicit anomaly and scenario annotations. Moreover, we design deceptive normal scenarios, where anomalous-looking metric patterns are explained by benign log events. These properties make CLOUDANOBENCH substantially more ambiguous and challenging than prior datasets.

To illustrate the utility of CLOUDANOBENCH, we propose CLOUDANOAGENT, an LLM-based agent framework enhanced with symbolic verification that jointly leverages both metrics and logs. Traditional approaches are often limited in handling unstructured log data, which contains critical temporal and behavioral context (Abdallah et al., 2024; Lou et al., 2019). CLOUDANOAGENT integrates a Fast and Slow Detection mechanism, leveraging the reasoning capabilities of LLMs to analyze metrics alongside contextual event logs, thereby ensuring both responsiveness and accuracy in detection. Furthermore, we design an symbolic verification module (Sun & Bookman, 1994), which performs statistical validation over metric patterns and regex-based matching over log events for each anomaly scenario. Acting as a critic to the *Integrated Agent*, this component further improves detection performance and reduces false positives. As a result, CLOUDANOAGENT demonstrates strong performance in both anomaly detection and scenario identification, while also exhibiting generalizability beyond the capabilities of existing methods.

We evaluate CLOUDANOBENCH and CLOUDANOAGENT through extensive experiments. CLOUDANOAGENT achieves up to nearly 20% higher F1-score on anomaly detection and 15% on scenario identification over baselines (e.g., machine learning methods and vanilla LLM prompting), which often struggle with deceptive normal cases and accurate scenario identification. Notably, CLOUDANOAGENT also shows competitive performance on log-only, point-anomaly datasets, highlighting its strong generalizability beyond its original design for context-level multimodal anomalies.

Our contributions can be summarized as:

- We introduce CLOUDANOBENCH, a large-scale benchmark for context anomalies that jointly includes metrics and logs, designed to evaluate both anomaly detection and scenario identification.
- We propose CLOUDANOAGENT, an LLM-based agent framework guided by symbolic verification, which simultaneously leverages metrics and logs and exhibits strong generalization ability.
- Experimental results demonstrate the challenging and deceptive nature of CLOUDANOBENCH, as well as the superior performance of CLOUDANOAGENT in both anomaly detection and scenario identification, achieving higher performance than compared methods and strong generalization on other datasets.

## 2 RELATED WORK

Anomaly detection is the process of identifying and extracting unexpected behaviors and patterns from the data. As cloud systems continue to evolve and grow rapidly, detecting anomalies has become critical to ensure cloud stability and reliability. Early deep learning approaches like LogAnomaly (Meng et al., 2019) used LSTMs to model normal sequences of log keys and detect deviations. This was advanced by LogBERT (Guo et al., 2021), which adapted Transformer to learn deep contextual representations of logs through self-supervised pre-training. More recently, LogLLM (Guan et al., 2024) has demonstrated the power of LLMs, coordinating BERT and Llama to extract semantic vectors and classify log sequences. These methods are designed specifically for log-based point anomalies. Therefore, we use them as baselines for evaluation in this work to demonstrate CLOUDANOAGENT's performance when the data is limited to log only.

---

[1]The current benchmark is available at https://anonymous.4open.science/r/CloudAnoBench-17ED

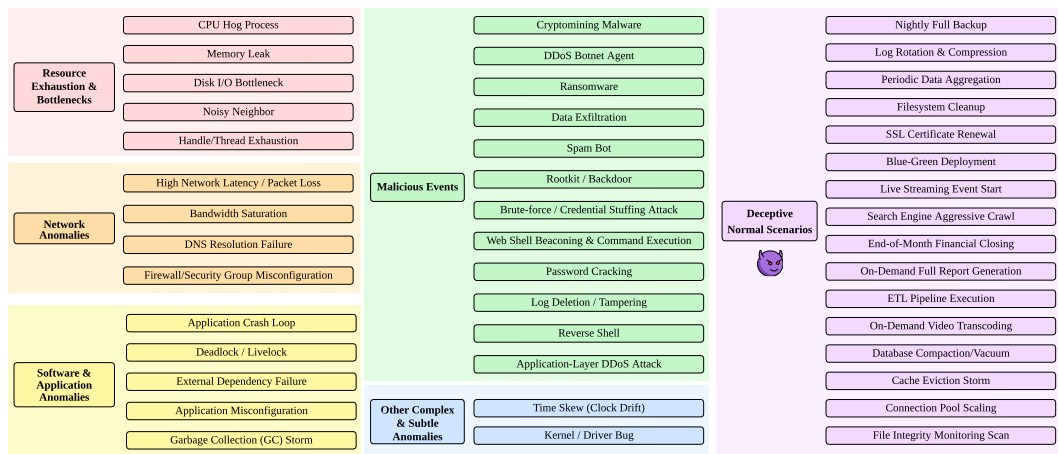

Figure 1: Overview Taxonomy of CloudAnoBench

TraceBench (Zhou et al., 2014) is a dataset of trace anomalies collected from real-world distributed systems, containing 370,000 traces and covering 17 types of anomalies. HPC dataset (Makanju et al., 2009) offers log-based context anomalies collected from a high performance computing cluster. However, it does not contain metrics data and its data are not labeled, limiting its applicability for evaluating techniques. IBM Cloud Console (Islam et al., 2025) provides context anomaly data from cloud systems that are properly labeled. However, this dataset contains metrics data only. More recently, LO2 (Bakhtin et al., 2025) is a novel comprehensive dataset of logs, metrics, and traces from a microservice system, encompassing multiple data modalities. Although not specific to cloud environments, it highlights the significance of using contextual data for effective anomaly detection.

## 3 CLOUDANOBENCH

| Dataset | Metrics & Logs | Type | Labeled | # Scenarios |
|---|---|---|---|---|
| BGL (Oliner & Stearley, 2007) | ✗ | Point | ✓ | – |
| Thunderbird (Oliner & Stearley, 2007) | ✗ | Point | ✓ | – |
| HPC (Makanju et al., 2009) | ✗ | Context | ✗ | – |
| HDFS_v1 (Xu et al., 2009) | ✗ | Point | ✓ | 17 |
| BETH (Highnam et al., 2021) | ✗ | Point | ✓ | – |
| IBM Cloud Console (Islam et al., 2025) | ✗ | Context | ✓ | – |
| RS-Anomic (Akmeemana et al., 2025) | ✗ | Point | ✓ | 10 |
| **CloudAnoBench (Ours)** | ✓ | Context | ✓ (scenarios) | 28 |

Table 1: Comparison of CloudAnoBench with existing datasets, highlighting its various scenarios, clear annotations with scenarios, and coverage of context anomalies that jointly include both metrics and logs.

We introduce CLOUDANOBENCH, a large-scale benchmark for evaluating context-aware anomaly detection in cloud environments. CLOUDANOBENCH consists of 1,252 cases covering 28 anomalous scenarios and 16 normal scenarios, as presented in Figure 1, totaling approximately 200,000 lines of data. Unlike existing benchmarks, shown in Table 1, it jointly includes context-level metrics and log data, with each case annotated by explicit anomaly and scenario labels. Moreover, we emphasize that anomaly detection methods should not raise alerts under normal operating conditions, as false alarms can incur significant financial and operational costs. To this end, CLOUDANOBENCH also incorporates a diverse set of normal scenarios that occur frequently in practice yet remain highly challenging for detectors. These properties make CLOUDANOBENCH substantially more challenging for prior machine learning and LLM prompting approaches, while also covering a broader spectrum of anomaly scenarios and aligning more closely with real-world cloud operations. More details

**Algorithm 1:** CLOUDANOBENCH Generation with Manual Review

**Input:** Scenario set $\mathcal{S}$ with anomaly and normal cases, example $e$
**Output:** Final dataset $\mathcal{D}$ with paired metrics.csv and log files

1 **foreach** *scenario* $s \in \mathcal{S}$ **do**
2  **Action:** Construct structured prompt $P_s$
3  **if** *example $e$ exists for $s$* **then**
4   Attach $e$ as one-shot reference to guide consistency in format and content;
5  **end**
6  **for** $i \leftarrow 1$ **to** *number of required cases* **do**
7   (Generate Metrics Data)
8   $f_{\text{metrics}} \leftarrow$ LLM.GenerateCode($P_s$) with `code_execution`;
9   metrics.csv $\leftarrow$ Execute($f_{\text{metrics}}$);
10   **Action:** anomaly pattern and data format verification
11   (Generate Log Data)
12   log $\leftarrow$ LLM.GenerateLog($P_s$, aligned with metrics.csv);
13   **Action:** alignment with metrics, avoid metric description, insert benign noise
14   (Verification and Manual Review)
15   **if** *GPT-4o.Verifier($metrics, log$)=pass* **and** *ManualReview=pass* **then**
16    Append ($metrics, log, label$) to $\mathcal{D}$;
17   **end**
18   **else**
19    Request regeneration of the case;
20   **end**
21  **end**
22 **end**
23 **return** $\mathcal{D}$;

of CLOUDANOBENCH are provided in Appendix D and Appendix E, while additional information about the compared datasets is presented in Appendix G.

## 3.1 SCENARIO EXTRACTION

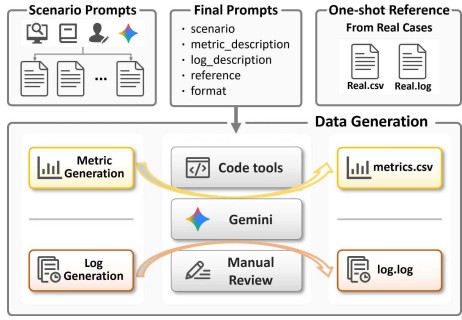

Figure 2: CloudAnoBench Construction

We categorize the anomaly scenarios covered in prior work (Garraghan et al., 2014; Bodik et al., 2010; Gunawi et al., 2014) into five major classes: **Resource Exhaustion & Bottlenecks**, **Network Anomalies**, **Software & Application Anomalies**, **Malicious Events**, and **Other Complex & Subtle Anomalies**. Notably, CLOUDANOBENCH encompasses not only anomaly scenarios triggered by genuine system faults but also deceptive normal scenarios, where anomalous-looking metric patterns are explained by benign log events. Such nuanced scenarios closely mirror real-world conditions, where misleading signals often lead traditional detectors to false positives. Each case is systematically documented with its anomaly category, anomaly scenario (if applicable), description, and corresponding metrics and logs. To ensure clarity and consistency across the dataset, instead of retaining the original heterogeneous formats, we employ Gemini-2.5-Flash to standardize every case into a structured prompt format, thereby facilitating reproducibility and enabling further data generation.

## 3.2 DATA GENERATION

To construct CLOUDANOBENCH, we leverage Gemini-2.5-Pro with scenario-specific prompts and tool invocation. Since many anomalies (e.g., cryptomining) cannot be safely reproduced on real

servers, and prior studies have demonstrated the feasibility of LLM-based dataset construction, we adopt an LLM-assisted pipeline. One real-world examples are provided as one-shot references to stabilize generation and ensure consistency. Each case consists of synchronized `metrics.csv` and `log` files that simulate realistic cloud environments. The full process, including prompt construction, metric/log generation, verification, and manual review, is summarized in Algorithm 1.

**Metrics Data.** Each case contains multivariate system metrics (CPU, GPU, memory, disk I/O, and network), spanning 90 lines over 450 seconds. Five canonical anomaly patterns are simulated (spike, dip, gradual increase, gradual decrease, fluctuation), with values sampled under realistic constraints to ensure plausibility. Metric data is generated via LLM with `code_execution`, which produces valid `.csv` files.

**Log Data.** Log data spans the same 450-second window, with 40–80 entries temporally aligned to the metrics. Logs are generated to avoid direct metric restatements while incorporating benign noise (e.g., SSH logins, scheduled tasks), thereby reflecting real-world conditions.

### 3.3 MANUAL REVIEW WITH LLM

To ensure data quality, we combine automatic checks with GPT-4o verification and human review. GPT-4o filters low-quality or inconsistent cases, while human reviewer adjudicate flagged samples and confirm anomaly classifications, ensuring realism and consistency across modalities.

As a result, CLOUDANOBENCH offers a large-scale, systematically validated, and multimodal benchmark that faithfully reflects realistic cloud anomalies while maintaining consistency, diversity, and reproducibility. Examples can be found in Appendix F. To verify that CLOUDANOBENCH reflects real-world data patterns, we conducted an additional comparison experiment using the RS-Anomic dataset (Akmeemana et al., 2025). RS-Anomic provides multivariate metrics collected from the RobotShop Instana microservice system and is widely used for anomaly detection and root-cause analysis in production-like settings. The experiment evaluates the similarity between the temporal behaviors in CLOUDANOBENCH and those in RS-Anomic. Full results are reported in Appendix C.

## 4 CLOUDANOAGENT

In this work, we propose CLOUDANOAGENT, an LLM-based agent guided by symbolic verification for cloud anomaly detection. **The main innovation lies in proposing a novel context engineering framework specifically for cloud anomaly detection.** Specifically, Our framework is comprised of two main modules: Fast and Slow Detection and Symbolic Verifier. Fast and Slow Detection consists of three agents: a Metrics Agent, a Log Agent, and an Integrated Agent that jointly produce an initial decision. This initial decision is then validated by a Symbolic Verifier module, which applies rigorous rule-based symbolic techniques to ensure reliability. This validation also helps to refine the hypotheses for the Integrated Agent, effectively creating a feedback loop between the two modules. This hybrid approach significantly enhances detection precision and reduces false positives.

### 4.1 FAST AND SLOW DETECTION

To capture both short-term anomalies (e.g., sudden spikes or drops) and context-dependent anomalies (e.g., deviations explainable only through associated log events), CLOUDANOAGENT adopts a Fast and Slow Detection mechanism. This dual-phase strategy enables (1) responsiveness, by performing real-time detection of abrupt anomalous signals, (2) interpretability, by uncovering underlying causes and temporally extended anomaly patterns through deeper log reasoning, and (3) precision, by reducing false positives through joint reasoning. Both Fast and Slow Detection are performed by LLM-based agents, yet they diverge in terms of their invocation timing, agent composition, and, most importantly, the design of their respective context engineering workflows.

**Fast Detection** is performed by a dedicated Metrics Agent, designed to continuously monitor real-time system signals within a sliding time window. This agent summarizes the behavior of metrics and outputs a structured report indicating whether an anomaly was detected and which metrics were affected, thereby providing the responsiveness needed for timely intervention.

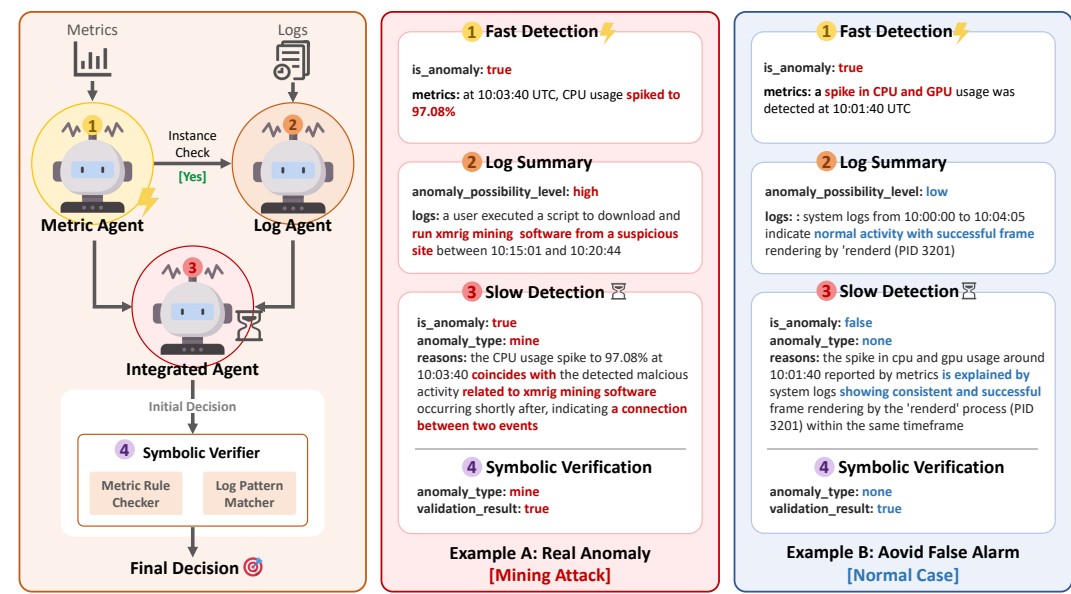

Figure 3: Overview of CloudAnoAgent. The red and blue boxes illustrate example outputs of CloudAnoAgent for different cases (simplified version).

**Slow Detection** is triggered in an event-driven manner upon anomaly signals raised by the Metrics Agent. Prior studies have shown that metric-based methods often suffer from high false positive rates in real-world cloud systems due to the lack of contextual information (Barbhuiya et al., 2018b; Kumari et al., 2024). To address this, our approach complements time-series signals with log-based reasoning to determine whether the detected anomaly reflects a truly abnormal event or a benign fluctuation. Beyond binary validation, this phase also identifies the likely root cause and higher-level anomaly event behind the observed deviations.

Specifically, a Log Agent processes log entries temporally aligned with the anomalous window, extracting the semantic meaning of system behaviors and assigning an anomaly likelihood level (i.e., low, medium, or high). These outputs, together with the response from the Metrics Agent, are fed into the Integrated Agent, which integrates both perspectives to perform joint anomaly verification and fine-grained classification. It determines whether the anomaly is valid and, if so, outputs the corresponding anomaly scenario along with a natural language explanation. This completes the end-to-end reasoning cycle in CloudAnoAgent's Fast and Slow Detection mechanism.

## 4.2 SYMBOLIC VERIFIER

Despite the improved performance of our Fast and Slow Detection over other methods in terms of accuracy and false positive rate, the inherent stochasticity of LLM outputs raises concerns about reliability. The symbolic verifier is introduced because even with the fast–slow detection pipeline, LLM outputs can still exhibit stochastic errors, leading to occasional misclassification in highly deceptive cases. Its role is to provide a stable layer of constraint checking that prevents major category-level mistakes rather than determining fine-grained scenarios.

To address this, we introduce a symbolic verifier that not only validates the Integrated Agent's initial decisions but also helps the system reflect and refine them when inconsistencies arise.

$$\text{Verifier}(M, L, s) = \begin{cases} 1, & \text{if VerifyMetric}(M, s) = 1 \ \land \ \text{VerifyLog}(L, s) = 1, \\ 0, & \text{otherwise.} \end{cases} \quad (1)$$

Specifically, for each predefined anomaly scenario $s \in \mathcal{S}$, the symbolic verifier applies two complementary checks: (1) a metric rule checker, which evaluates whether the observed metric sequence $M$ satisfies the expected statistical signature of $s$, and (2) a log pattern matcher, which uses regex-based

keyword matching to confirm whether the log entries $L$ contain events consistent with $s$. Formally, let VerifyMetric$(M, s) \in \{0, 1\}$ denote the binary outcome of metric validation for scenario $s$, and VerifyLog$(L, s) \in \{0, 1\}$ the corresponding outcome for log validation. The symbolic verifier accepts scenario $s$ if and only if both conditions hold:

For the detector's predicted scenario $\hat{s}$, if Verifier$(M, L, \hat{s}) = 1$, the decision is confirmed; otherwise, the system iteratively evaluates all $s \in \mathcal{S}$ and either converges to a consistent alternative or abstains after a maximum retry limit. Appendix J provides an simplified example for the mining scenario. For rule construction, we use static, non-learned rules: an LLM first generates candidate metric and log rules, and human experts then review them for correctness and generality. Crucially, reviewers do not see CloudAnoBench's data distribution, ensuring the rules remain dataset-agnostic and avoid overfitting.

## 5 EXPERIMENTS

This section shows our experimental setup, including datasets, compared methods, evaluation metrics, and implementation details. We evaluate detection methods on precision, recall, false positive rate (FPR) and F1-score on anomaly detection and accuracy on scenario identification (SI).

### 5.1 EXPERIMENTAL SETTINGS

#### 5.1.1 DATASETS

We conducted extensive experiments on four datasets to evaluate the effectiveness of the proposed methods for anomaly detection in cloud environments. These datasets include **BGL**, **Thunderbird**, **HDFS_v1**, and our proposed benchmark, CLOUDANOBENCH. Table 1 provides an overview of these datasets. Although the IBM Cloud Console dataset also contains context-level anomalies, it is excluded from our evaluation because its anomalies are labeled using IBM's proprietary internal monitoring tools, making it overly enterprise-specific and less suitable for general benchmarking (Islam et al., 2021). The use of BGL, Thunderbird, and HDFS_v1 further demonstrates that while CLOUDANOAGENT is primarily designed to address context-level anomalies involving both metrics and logs, it also achieves strong performance on point-level anomaly datasets.

#### 5.1.2 COMPARED METHODS

We compare our proposed CLOUDANOAGENT framework against three categories of baselines: machine learning (ML) methods, vanilla LLM prompting, and log-only detection methods.

- **Machine Learning Algorithms.** We include several widely used machine learning baselines, including tree-based models such as *Isolation Forest* and *Decision Tree*, linear models such as *Logistic Regression*, unsupervised clustering methods such as *K-Means* and the *Rarity Model*, as well as the *OOV Detector*, which flags infrequent or previously unseen log tokens. These methods rely on fixed statistical patterns or token rarity assumptions, provide efficient and lightweight detection, and have been applied to anomaly detection tasks in microservice APIs (Bakhtin et al., 2025).

- **Vanilla LLM Prompting** To isolate the effectiveness of CloudAnoAgent's agent architecture and symbolic verification from the raw capabilities of LLMs, we introduce a simple LLM baseline. In this setting, both the metrics data and the corresponding log text are directly fed into the LLM, which is prompted to output a binary anomaly decision along with a categorical scenario identification, without any agent decomposition or symbolic verification.

- **Log-Only Detection Methods.** We also compare against log-specific anomaly detection approaches, including LogAnomaly (Meng et al., 2019), LogLLM (Guan et al., 2024), and Log-BERT (Guo et al., 2021). These methods are designed primarily for point anomaly detection in logs and serve as strong baselines to evaluate how well CLOUDANOAGENT generalizes to log-only detection settings.

## 5.2 EVALUATION METRICS

In this paper, we evaluate the performance of the proposed methods on both **anomaly detection** and **scenario identification**. For anomaly detection, the task is modeled as a binary classification problem. We report Precision, Recall, FPR, and F1-score, which together capture the reliability of positive alerts, the ability to identify true anomalies, the tendency to produce false alarms, and the overall balance between precision and recall.

For scenario identification (SI), we formulate the task as a multi-class classification problem and evaluate performance using accuracy (Acc). Let $N$ denote the total number of cases, $y_i$ the ground-truth label of the $i$-th case, and $\hat{y}_i$ the predicted label. Accuracy is defined as:

$$\text{Acc} = \frac{|\{i \mid \hat{y}_i = y_i\}|}{N}, \tag{2}$$

where $|\{i \mid \hat{y}_i = y_i\}|$ denotes the number of correctly classified cases. In other words, accuracy directly measures the proportion of correctly identified anomaly scenarios in the benchmark.

## 5.3 IMPLEMENTATION DETAILS

For both CLOUDANOAGENT and the LLM baseline, we accessed LLMs via API calls using the default inference parameters provided by each vendor. Since ML methods cannot directly process textual log information, we add a log parser so that they can make predictions using both metrics and logs. Moreover, since the BGL, Thunderbird, and HDFS_v1 datasets only contain log data, the CLOUDANOAGENT was evaluated on these datasets without the metrics agent component. All experiments were conducted on a computing cluster equipped with two NVIDIA A100 GPUs. Some example prompts are provided in Appendix I.

## 6 RESULTS

| Setting | Model | Anomaly Detection | | | | SI |
|---|---|---|---|---|---|---|
| | | **P** | **R** | **FPR** | **F1** | **Acc** |
| **ML Algorithms** | IsolationForest | 54.0 | 98.1 | 100.0 | 69.7 | 53.5 |
| | KMeans | 55.3 | 93.0 | 90.0 | 69.4 | 55.2 |
| | RarityModel | 76.5 | 32.8 | 12.1 | 45.9 | 57.9 |
| | OOV Detector | 60.4 | 63.2 | 49.6 | 61.8 | 57.4 |
| | **Average** | 61.55 | 71.78 | 63.0 | 61.95 | 55.0 |
| **Vanilla LLM** | Gemini 2.5 Flash | 66.9 | 91.4 | 54.3 | 77.2 | 66.9 |
| | Gemini 2.5 Flash Lite | 62.4 | 89.0 | 64.5 | 73.3 | 61.2 |
| | Gemini 2.0 Flash Lite | 60.4 | 86.1 | 67.7 | 71.0 | 60.0 |
| | GPT-4o Mini | 58.9 | 83.6 | 69.9 | 69.1 | 68.8 |
| | GPT-4o | 66.7 | 88.4 | 52.9 | 76.1 | 58.1 |
| | Qwen Plus | 60.3 | 83.9 | 66.3 | 70.2 | 60.3 |
| | **Average** | 62.6 | 87.1 | 62.6 | 72.8 | 62.6 |
| **CloudAnoAgent w/o and with symbolic verifier** | Gemini 2.5 Flash | 86.9/90.0 | 94.3/96.6 | 17.0/12.8 | 90.4/93.2 | 73.8/78.6 |
| | Gemini 2.5 Flash Lite | 85.9/88.7 | 91.9/92.2 | 18.1/14.1 | 88.8/90.5 | 73.2/76.3 |
| | Gemini 2.0 Flash Lite | 81.9/87.9 | 90.8/90.5 | 24.1/14.9 | 86.1/89.2 | 70.7/74.5 |
| | GPT-4o Mini | 87.5/90.0 | 89.5/90.8 | 15.3/12.1 | 88.5/90.4 | 71.9/73.1 |
| | GPT-4o | 91.5/92.5 | 93.6/95.6 | 10.4/9.3 | 92.5/94.0 | 74.8/79.2 |
| | Qwen Plus | 87.1/88.7 | 92.2/92.7 | 16.3/14.2 | 89.6/90.6 | 74.5/76.3 |
| | **Average** | 86.8/89.6 | 92.0/93.1 | 16.9/12.9 | 89.3/91.3 | 73.2/76.3 |

Table 2: Evaluation results on CloudAnoBench: Precision (P), Recall (R), False Positive Rate (FPR), and F1-score for anomaly detection, and Accuracy (Acc) for scenario identification (SI).

This section presents our results on both anomaly detection and anomaly scenario identification. We analyze how well different methods detect anomalies, challenges of identifying specific anomaly scenarios, and evaluate the generalization performance of CLOUDANOAGENT across diverse datasets and anomaly types.

## 6.1 ANOMALY DETECTION PERFORMANCE

Our evaluation on anomaly detection (Table 2) demonstrates that CLOUDANOAGENT consistently outperforms all baselines on CLOUDANOBENCH, achieving the highest average F1-score (91.3%) and the lowest average FPR (12.9%). These results highlight its ability to capture true anomalies while suppressing false alarms, even in deliberately misleading normal cases. In contrast, vanilla LLM prompting attains competitive Recall (87.1% on average) but suffers from extremely high FPR (62.6% on average), indicating poor robustness against deceptive normal patterns and raising concerns about practical deployment costs. ML algorithms such as KMeans and RarityModel achieve recall above 95%, but exhibit poor precision and FPR. Finally, incorporating symbolic verification further improves CLOUDANOAGENT, reducing FPR by 4% and increasing F1 by 2% on average, underscoring the verifier's role as a reliable critic that enhances detection accuracy. More ablation study results can be found in Appendix B.

## 6.2 CHALLENGES IN SCENARIO IDENTIFICATION

| CloudAnoAgent | Number of Scenarios | | |
| --- | --- | --- | --- |
| Model | 5 | 10 | 29 |
| Gemini 2.5 Flash | 100 | 98.25 | 78.62 |
| Gemini 2.5 Flash Lite | 100 | 96.49 | 76.28 |
| Gemini 2.0 Flash Lite | 97.35 | 91.23 | 74.52 |

Table 3: Performance of CLOUDANOAGENT across different numbers of anomaly scenarios.

Anomaly scenario identification is an important yet often overlooked task. Accurate identification not only assists engineers, but also enables future server-side intelligent agents to more quickly trace the root causes of anomalies and accelerate recovery. Table 2 shows that CLOUDANOAGENT achieves, on average, a 13.7% improvement over vanilla LLM prompting and a 20% improvement over ML algorithms. However, as shown in Table 3, as the number of anomaly scenarios increases, scenario identification accuracy decreases significantly. This is because more scenarios require longer context windows and introduce higher semantic similarity across scenarios, making fine-grained identification more challenging for LLM-based agents.

## 6.3 GENERALIZATION OF CLOUDANOAGENT

| | HDFS_v1 | | | ThunderBird | | | BGL | | |
| --- | --- | --- | --- | --- | --- | --- | --- | --- | --- |
| Model | Prec. | Rec. | F1 | Prec. | Rec. | F1 | Prec. | Rec. | F1 |
| LogLLM | 99.4 | 100.0 | 99.7 | 96.6 | 96.6 | 96.6 | 86.1 | 97.9 | 91.6 |
| LogAnomaly | 96.0 | 94.0 | 95.0 | - | - | - | 97.0 | 94.0 | 96.0 |
| LogBERT | 87.0 | 78.1 | 82.3 | 96.8 | 96.5 | 96.6 | 89.4 | 92.3 | 90.8 |
| **CloudAnoAgent** | 89.5 | 95.1 | 92.2 | 89.6 | 96.4 | 92.9 | 98.5 | 99.7 | 99.1 |

Table 4: Performance comparison on three log-only datasets with point anomaly type.

It is worth noting that while CLOUDANOAGENT is primarily designed for context-level anomaly datasets that include both metrics and logs, it can still be adapted to different data modalities and anomaly types through simple modifications to the prompting rules and lightweight adjustments to the agent workflow. As shown in Table 4, CLOUDANOAGENT achieves competitive performance on HDFS_v1, ThunderBird, and BGL, when compared to LogLLM, LogAnomaly, and Log-BERT—methods that are specifically designed for log-based point anomaly detection. The results of these baseline methods on the three datasets are reported from each original paper, and implementation details can be found in Appendix H.

## 6.4 LATENCY PERFORMANCE

In cloud environments, latency is an important consideration for anomaly detection. While CloudAnoAgent does exhibit higher detection latency than traditional ML baselines, this difference reflects a fundamental performance–latency trade-off. Classical ML models provide very low latency but suffer from high false positive rates, which in real deployments lead to operational overhead, alert fatigue, and inefficient use of engineering time. By contrast, CloudAnoAgent achieves

| Model | Mean ± Std Dev | Max |
|---|---|---|
| IsolationForest | 0.61 ms ± 0.59 ms | 4.31 ms |
| KMeans | 0.25 ms ± 0.46 ms | 5.54 ms |
| RarityModel | 1.59 ms ± 0.78 ms | 12.76 ms |
| OOV Detector | 1.15 ms ± 0.68 ms | 19.07 ms |

Table 5: Latency of ML Algorithms

| Model | Total Latency |
|---|---|
| Gemini 2.5 Flash | 7.514 s |
| Gemini 2.5 Flash Lite | 0.577 s |
| Gemini 2.0 Flash | 0.671 s |
| GPT-4o | 2.917 s |
| GPT-4o Mini | 1.834 s |
| Qwen Plus | 2.761 s |

Table 6: Latency of Vanilla Prompting

| Model | Metrics Agent | Log Agent | Integrated Agent | Symbolic Verifier | Total |
|---|---|---|---|---|---|
| Gemini 2.5 Flash | 3.78 s | 2.91 s | 3.45 s | 0.42 s | 10.56 s |
| Gemini 2.5 Flash Lite | 0.63 s | 0.59 s | 0.71 s | 0.42 s | 2.35 s |
| Gemini 2.0 Flash | 1.08 s | 0.82 s | 1.05 s | 0.39 s | 3.34 s |
| GPT-4o | 2.33 s | 1.92 s | 2.09 s | 0.41 s | 6.75 s |
| GPT-4o Mini | 1.35 s | 1.09 s | 1.23 s | 0.40 s | 4.07 s |
| Qwen Plus | 5.67 s | 2.75 s | 5.29 s | 0.42 s | 14.14 s |

Table 7: Latency of CloudAnoAgent

substantially lower false positive rates and can identify the specific anomaly scenario, enabling faster and more accurate root-cause analysis. Additionally, end-to-end latency is not fixed; it varies significantly depending on factors such as API provider, model size, and whether the LLM is locally hosted. Using smaller locally deployed models can further reduce latency when needed.

## 7 CONCLUSION AND FUTURE WORK

In this work, we introduce CLOUDANOBENCH, a large-scale benchmark for context-level anomaly detection in cloud environments. Unlike existing datasets, our benchmark jointly incorporates both metrics and logs, and provides annotations not only for anomaly presence but also for specific anomaly scenarios. It covers 29 anomalous scenarios and further includes challenging deceptive normal cases designed to test the false positive robustness of detection methods. To address these challenges, we propose CLOUDANOAGENT, a symbolic-verification guided LLM-based multi-agent system capable of handling multimodal data for anomaly detection. Experimental results demonstrate that CLOUDANOAGENT significantly outperforms baselines in both anomaly detection and anomaly scenario identification, while also showing promising generalization to point anomaly datasets of different modalities. Future work will explore advanced post-training strategies and LLM architectural improvements to further enhance scenario identification accuracy, as well as optimizations to reduce detection latency in real-world deployments.

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

# A LIMITATIONS

Because CLOUDANOAGENT incorporates LLMs in both anomaly detection and scenario reasoning, its performance is not entirely deterministic. Subtle differences in prompt wording or changes in model versions may lead to slightly different outputs, introducing variability across repeated experiments. Similarly, since CLOUDANOBENCH contains partially LLM-synthesized data, the dataset itself may be sensitive to model behavior and prompt design. While we carefully design prompts and incorporate symbolic checks to mitigate these issues, achieving strict reproducibility under evolving LLM behavior remains challenging.

# B ABLATION STUDY FOR METRICS AGENT ONLY AND LOG AGENT ONLY

| Model | P | R | FPR | F1 | SI:Acc |
|---|---|---|---|---|---|
| Gemini 2.5 Flash | 58.55 | 93.27 | 79.26 | 71.94 | 0.426 |
| Gemini 2.5 Flash Lite | 59.31 | 93.70 | 77.15 | 72.64 | 0.407 |
| Gemini 2.0 Flash Lite | 57.57 | 87.99 | 77.86 | 69.60 | 0.353 |
| GPT-4o Mini | 57.14 | 86.68 | 78.03 | 68.88 | 0.387 |
| GPT-4o | 59.04 | 89.90 | 74.87 | 71.27 | 0.411 |
| Qwen Plus | 59.10 | 90.34 | 75.04 | 71.45 | 0.382 |

Table 8: Ablation Study: CloudAnoAgent (Metrics Agent Only)

| Model | P | R | FPR | F1 | SI:Acc |
|---|---|---|---|---|---|
| Gemini 2.5 Flash | 76.28 | 85.21 | 31.81 | 80.50 | 0.543 |
| Gemini 2.5 Flash Lite | 76.93 | 84.48 | 30.40 | 80.53 | 0.518 |
| Gemini 2.0 Flash Lite | 78.55 | 80.97 | 26.54 | 79.74 | 0.452 |
| GPT-4o Mini | 78.68 | 80.53 | 26.19 | 79.59 | 0.539 |
| GPT-4o | 78.17 | 83.89 | 28.12 | 80.93 | 0.574 |
| Qwen Plus | 78.13 | 82.14 | 27.59 | 80.09 | 0.553 |

Table 9: Ablation Study: CloudAnoAgent (Log Agent Only)

# C ALIGNMENT BETWEEN CLOUDANOBENCH AND REAL-WORLD DISTRIBUTION

To examine whether CLOUDANOBENCH reflects realistic temporal behaviors rather than synthetic artifacts, we performed an alignment study comparing our benchmark with the RS-Anomic dataset (Akmeemana et al., 2025). RS-Anomic contains multivariate system metrics collected from the RobotShop Instana microservice application and is widely used for anomaly detection and root-cause localization in production-like environments.

Since direct comparison of logs is not meaningful due to system-specific identifiers and environment-dependent log formats, our analysis focuses on temporal similarity between system metrics. We selected two anomaly categories in CLOUDANOBENCH (CPU and networking) and matched them with their closest counterparts in RS-Anomic.

## C.1 SIMILARITY METRICS

We applied three complementary sequence-similarity metrics:

- **Spearman's** $\rho$: measures similarity in monotonic trends.
- **Kendall's** $\tau$: evaluates consistency in pairwise rank ordering.
- **DTW/len**: normalized Dynamic Time Warping distance, which quantifies the temporal deformation needed to align two sequences.

These metrics jointly capture trend agreement, local ordering consistency, and temporal-shape similarity.

| Category | Scenario | Spearman $\rho$ | Kendall $\tau$ | DTW/len |
|---|---|---|---|---|
| CPU | High CPU Usage / CPU Hog | 0.816 | 0.656 | 0.179 |
| Network | High Latency | 0.688 | 0.513 | 0.169 |
| Network | Out-of-order Packets | 0.659 | 0.496 | 0.160 |
| Network | Packet Loss | 0.659 | 0.494 | 0.182 |

Table 10: Similarity between CLOUDANOBENCH and RS-Anomic metric traces across matched anomaly scenarios.

## C.2 FINDINGS

Across CPU and multiple network anomalies, we observe consistently strong alignment:

- **CPU anomalies** show high agreement in global trends and temporal patterns ($\rho = 0.816$, $\tau = 0.656$, DTW/len $= 0.179$).

- **Network anomalies** also demonstrate close correspondence to real-world dynamics, with $\rho \approx 0.66$–$0.69$ and DTW/len $\approx 0.16$–$0.18$.

These quantitative results are supported by qualitative inspection. For matched scenarios, both datasets show characteristic rise–fall dynamics, temporal jitter patterns, and noise structure. CPU hog events display rapid saturation and sustained utilization, while network-related anomalies exhibit latency buildup, irregular spikes, and transient bursts.

Although CLOUDANOBENCH is constructed using LLM-assisted scenario design and code-driven data generation, the resulting metric traces follow real-world temporal signatures closely. The strong correspondence observed across both quantitative metrics and qualitative patterns supports the realism of CLOUDANOBENCH and reinforces the validity of evaluating CloudAnoAgent on this benchmark.

| Scenario | Metrics | Logs |
| --- | --- | --- |
| CPU Hog Process | CPU spike | Timeout errors |
| Memory Leak | CPU fluctuation, Memory increase, Disk I/O spike | OOM killer |
| Disk I/O Bottleneck | CPU iowait spike, Disk I/O saturation, Network dip/fluctuation | Task blocked, Slow queries |
| Noisy Neighbor | CPU steal spike | Unexplained slowdowns, Timeouts |
| Handle/Thread Exhaustion | CPU & Memory spike, Network dip | Resource allocation errors |
| High Network Latency / Packet Loss | CPU spike, Memory increase, Network dip | Connection errors |
| Bandwidth Saturation | CPU spike, Network cap plateau | Timeout errors, Packet loss |
| DNS Resolution Failure | Network drop | DNS resolution errors |
| Firewall/Security Group Misconfiguration | Network drop (blocked ports) | Blocked port timeouts |
| Application Crash Loop | CPU & Memory fluctuation, I/O fluctuation | Service crash/restart, Fatal errors |
| Deadlock / Livelock | CPU dip/spike, Throughput drop | Deadlock detected |
| External Dependency Failure | CPU dip/spike, Memory increase, Network dip | Dependency failures, 503 errors |
| Application Misconfiguration | CPU & Memory spike, Network dip, Fluctuations | Misconfiguration errors |
| Garbage Collection (GC) Storm | CPU spike, Memory sawtooth fluctuation, Throughput dip | GC pauses, Timeouts |
| Cryptomining Malware | CPU & GPU spike, Network fluctuation | No matching logs, Unknown process |
| DDoS Botnet Agent | CPU & Memory spike, Network saturation | Resource contention, Outbound flood |
| Ransomware | CPU & Memory spike, Disk I/O spike | File errors, Log clearing, Ransom note |
| Data Exfiltration | CPU & Memory & Disk I/O & Network spike | Abnormal queries, File reads, Data exfiltration |
| Spam Bot | CPU & Network spike | Spam activity, Bounce messages |
| Rootkit / Backdoor | N/A | Log tampering, Rootkit traces |
| Brute-force / Credential Stuffing Attack | CPU & Network spike | Authentication failures, Log storm |
| Web Shell Beaconing & Command Execution | CPU & Disk I/O spike | Suspicious repeated requests; unexpected process creation |
| Password Cracking | CPU & GPU spike, Memory spike, Disk I/O startup spike | Cracking process detected; bash/auth log evidence |
| Log Deletion / Tampering | CPU spike, Disk write spike | Logs removed/cleared; file-integrity alerts; possible bash history |
| Reverse Shell | CPU spikes, Persistent outbound connection | Unexpected outbound process; nonstandard outbound TCP in firewall/audit logs |
| Application-Layer DDoS Attack | CPU & Memory & Disk I/O spike | Endpoint request flood; DB slow queries and timeouts |
| Time Skew (Clock Drift) | N/A | Clock skew errors |
| Kernel / Driver Bug | Unpredictable spikes/fluctuations or drop to zero | Kernel panic, Segfaults, HW faults |

Table 11: Anomaly scenarios and their performance in metrics and logs.

# D ANOMALY SCENARIOS DETAILS

| Anomaly Scenario | Description |
| --- | --- |
| CPU Hog Process | A single or few processes continuously occupy extremely high CPU due to bugs or malicious behavior |
| Memory Leak | The application fails to release memory due to bugs, causing continuous memory depletion |
| Disk I/O Bottleneck | Disk read and write requests exceed hardware capacity |
| Noisy Neighbor | Other VMs on the same physical host overconsume shared resources in a virtualized environment |
| Handle/Thread Exhaustion | Processes open too many file handles or threads, reaching the system limit |
| High Network Latency / Packet Loss | Network congestion or failures cause packet delay or loss |
| Bandwidth Saturation | Outbound or inbound traffic reaches the cloud provider's bandwidth cap |
| DNS Resolution Failure | Domain names cannot be resolved to IP addresses |
| Firewall/Security Group Misconfiguration | Incorrect firewall or cloud security group rules block normal port connections |
| Application Crash Loop | Applications repeatedly crash due to fatal errors and are restarted by a watchdog |
| Deadlock / Livelock | Two or more threads/processes wait for each other's resources, halting progress (deadlock/livelock) |
| External Dependency Failure | Dependent databases, caches, or third-party APIs become unavailable |
| Application Misconfiguration | An incorrect configuration file causes app startup failures, functional errors, or performance degradation |
| Garbage Collection (GC) Storm | Excessive or long garbage collection pauses in managed-language apps (e.g., Java) |
| Cryptomining Malware | Attacker runs malicious program on the server to mine cryptocurrency (uses CPU/GPU). |
| DDoS Botnet Agent | Server used as a "bot" in a DDoS campaign, sending large volumes of attack traffic. |
| Ransomware | Ransomware encrypts files on the server and demands ransom. |
| Data Exfiltration | Attacker exfiltrates sensitive data from the server to an external host. |
| Spam Bot | Server used to send large volumes of spam or phishing emails. |
| Rootkit / Backdoor | Attacker installs stealthy persistent backdoor/rootkit to maintain covert access. |
| Brute-force / Credential Stuffing Attack | Ongoing automated credential-stuffing / brute-force attack against authentication services (SSH/RDP/Web). |
| Web Shell Beaconing & Command Execution | Web shell on server beaconing to C2 and executing commands |
| Password Cracking | Offline password cracking with john/hashcat |
| Log Deletion / Tampering | Log cleaning to cover tracks |
| Reverse Shell | Reverse shell established to attacker host |
| Application-Layer DDoS Attack | Application-level DoS via expensive functions |
| Time Skew (Clock Drift) | Server system time drifts from NTP standard |
| Kernel / Driver Bug | OS kernel or hardware driver bug is triggered, causing instability or crashes |

Table 12: Anomaly scenarios with descriptions

# E DECEPTIVE NORMAL SCENARIOS DETAILS

| Normal Scenario | Description |
| --- | --- |
| Nightly Full Backup | Nightly full backup with database dump and remote upload |
| Log Rotation & Compression | Scheduled log rotation and compression |
| Periodic Data Aggregation | Periodic data aggregation for reports |
| Filesystem Cleanup | Filesystem cleanup of expired files |
| SSL Certificate Renewal | Automatic SSL certificate renewal |
| Blue-Green Deployment | Blue-green deployment traffic switch |
| Live Streaming Event Start | Start of a large-scale live streaming event |
| Search Engine Aggressive Crawl | Aggressive crawl by search engine bots |
| End-of-Month Financial Closing | End-of-month financial closing workload |
| On-Demand Full Report Generation | On-demand full report generation |
| ETL Pipeline Execution | ETL pipeline execution with data extraction and loading |
| On-Demand Video Transcoding | On-demand video transcoding with FFmpeg |
| Database Compaction/Vacuum | Database compaction or vacuuming |
| Cache Eviction Storm | Cache eviction storm with many misses |
| Connection Pool Scaling | Dynamic expansion of DB connection pool |
| File Integrity Monitoring Scan | File integrity monitoring scan |

Table 14: Normal scenarios with descriptions.

| Normal Scenario | Metrics | Logs |
|---|---|---|
| Nightly Full Backup | CPU spike (compression), Disk I/O spike (read), Net out spike (upload) | Backup task logs |
| Log Rotation & Compression | CPU spike (compression), Disk I/O spike | Logrotate success logs |
| Periodic Data Aggregation | CPU spike (scheduled), Disk I/O spike | Report job logs, DB slow query logs |
| Filesystem Cleanup | CPU spike (scan), Disk I/O spike (read/write) | Cleanup script logs |
| SSL Certificate Renewal | Net in/out spike (CA communication), CPU spike (reload) | Certbot/ACME renewal logs |
| Blue-Green Deployment | Green env: CPU/Mem/Net spike; Blue env: CPU/Mem/Net dip | Load balancer & deployment logs |
| Live Streaming Event Start | Net out spike, Connection spike, CPU spike | Live stream logs, CDN access logs |
| Search Engine Aggressive Crawl | Net in/out spike, CPU spike, Disk I/O spike | Web access logs (bot user-agent) |
| End-of-Month Financial Closing | CPU/Mem/Disk I/O gradual increase & fluctuation | App logs with financial keywords |
| On-Demand Full Report Generation | CPU spike, Memory increase, Disk I/O spike | Report generation logs |
| ETL Pipeline Execution | CPU/Mem/Disk/Net spikes & fluctuations | ETL tool logs (Airflow/dbt etc.) |
| On-Demand Video Transcoding | Sustained CPU spike, Disk I/O spike | Transcoding job logs (FFmpeg) |
| Database Compaction/Vacuum | Disk I/O spike, CPU spike | DB compaction/vacuum logs |
| Cache Eviction Storm | DB CPU/Disk I/O/Net in spike, Cache misses spike | Cache miss logs, DB query surge logs |
| Connection Pool Scaling | App latency spike, CPU spike | Connection pool expansion logs |
| File Integrity Monitoring Scan | Disk I/O spike (read), CPU spike (hashing) | AIDE/Tripwire scan logs |

Table 13: Normal scenarios and their performance in metrics and logs.

# F  BENCHMARK EXAMPLES

| timestamp | cpu_usage | mem_usage | disk_io | net_in | net_out |
|---|---|---|---|---|---|
| 2025-07-12T04:00:00Z | 13.75 | 36.2 | 20.12 | 1.31 | 0.89 |
| 2025-07-12T04:00:05Z | 19.51 | 42.13 | 16.7 | 1.31 | 1.14 |
| 2025-07-12T04:00:10Z | 17.32 | 42.61 | 28.87 | 1.37 | 0.96 |
| 2025-07-12T04:00:15Z | 15.99 | 40.61 | 28.16 | 1.41 | 1.05 |
| 2025-07-12T04:00:20Z | 11.56 | 42.71 | 18.87 | 1.01 | 1.44 |
| 2025-07-12T04:00:25Z | 11.56 | 39.94 | 24.9 | 1 | 0.89 |
| 2025-07-12T04:00:30Z | 10.58 | 40.23 | 27.26 | 1.3 | 1.46 |
| 2025-07-12T04:00:35Z | 18.66 | 39.28 | 23.33 | 1.15 | 1.41 |
| 2025-07-12T04:00:40Z | 16.01 | 35.25 | 22.94 | 1.2 | 0.7 |
| 2025-07-12T04:00:45Z | 17.08 | 36.08 | 18.63 | 1.3 | 0.57 |
| 2025-07-12T04:00:50Z | 10.21 | 35.31 | 16.4 | 1.39 | 0.6 |
| 2025-07-12T04:00:55Z | 19.7 | 41.36 | 28.46 | 0.84 | 0.52 |
| 2025-07-12T04:01:00Z | 18.32 | 38.14 | 28.51 | 0.88 | 0.59 |
| 2025-07-12T04:01:05Z | 12.12 | 40.09 | 24.5 | 0.59 | 1.18 |
| 2025-07-12T04:01:10Z | 11.82 | 44.08 | 20.09 | 1.08 | 0.57 |
| 2025-07-12T04:01:15Z | 11.83 | 37.49 | 20.24 | 0.54 | 0.82 |
| 2025-07-12T04:01:20Z | 13.04 | 39.1 | 25.89 | 0.97 | 1.34 |
| 2025-07-12T04:01:25Z | 15.25 | 42.56 | 28.46 | 1.04 | 0.52 |
| 2025-07-12T04:01:30Z | 14.32 | 37.29 | 28.31 | 0.79 | 1.31 |
| 2025-07-12T04:01:35Z | 12.91 | 35.77 | 26.7 | 1.09 | 0.78 |
| 2025-07-12T04:01:40Z | 16.12 | 37.9 | 24.63 | 0.53 | 0.62 |
| 2025-07-12T04:01:45Z | 11.39 | 36.61 | 16.26 | 0.54 | 1.2 |
| 2025-07-12T04:01:50Z | 12.92 | 44.3 | 17.42 | 1.32 | 1.13 |
| 2025-07-12T04:01:55Z | 13.66 | 43.08 | 28.48 | 0.86 | 1.38 |
| 2025-07-12T04:02:00Z | 14.56 | 41.33 | 24.1 | 0.63 | 1.24 |
| 2025-07-12T04:02:05Z | 17.85 | 43.71 | 15.14 | 1.02 | 1.3 |
| 2025-07-12T04:02:10Z | 12 | 43.04 | 16.52 | 1.27 | 0.78 |
| 2025-07-12T04:02:15Z | 15.14 | 36.87 | 24.95 | 0.72 | 0.68 |
| 2025-07-12T04:02:20Z | 15.92 | 43.93 | 15.08 | 1.12 | 1.25 |
| 2025-07-12T04:02:25Z | 10.46 | 40.39 | 17.41 | 0.59 | 1.31 |

Figure 4: Metrics Example of Data Exfiltration (Simplified Version)

```
Jul 12 04:00:10 systemd[1]: Started Session 5 of user db_admin.
Jul 12 04:00:25 sshd[15482]: Accepted publickey for db_admin from 10.1.2.5 port 49822 ssh2
Jul 12 04:00:50 cron[11432]: (root) CMD (run-parts --report /etc/cron.hourly)
Jul 12 04:01:40 web-app[887]: INFO: User 'alex' successfully authenticated.
Jul 12 04:02:20 web-app[887]: GET /api/v1/dashboard HTTP/1.1 200
Jul 12 04:02:35 auditd[512]: type=SYSCALL msg=audit(...): arch=c000003e syscall=257 success=yes
exit=3 a0=ffffff9c a1=7ffd1f9b6e20 a2=90800 a3=0 items=1 ppid=15510 pid=15512 auid=1001 uid=1001
gid=1001 euid=1001 suid=1001 fsuid=1001 egid=1001 sgid=1001 fsgid=1001 tty=pts0 ses=5 comm="cat"
exe="/bin/cat" key="secret_file_access"
Jul 12 04:02:40 auditd[512]: type=PATH msg=audit(...): item=0 name="/etc/shadow" inode=131078
dev=fd:01 mode=0100640 ouid=0 ogid=42 rdev=00:00 nametype=NORMAL
Jul 12 04:02:50 auditd[512]: type=SYSCALL msg=audit(...): arch=c000003e syscall=257 success=yes
exit=4 a0=ffffff9c a1=7ffd1f9b6e20 a2=90800 a3=0 items=1 ppid=15510 pid=15515 auid=1001 uid=1001
gid=1001 euid=1001 suid=1001 fsuid=1001 egid=1001 sgid=1001 fsgid=1001 tty=pts0 ses=5 comm="cat"
exe="/bin/cat" key="secret_file_access"
Jul 12 04:02:55 auditd[512]: type=PATH msg=audit(...): item=0 name="/opt/app/credentials.env"
inode=262148 dev=fd:01 mode=0100600 ouid=1001 ogid=1001 rdev=00:00 nametype=NORMAL
Jul 12 04:03:10 mysqld[1234]: 2025-07-12T04:03:10.123456Z 115 [Note] User 'db_admin'@'localhost'
executed: 'SELECT * FROM customers.credit_card_details;'
Jul 12 04:03:30 mysqld[1234]: 2025-07-12T04:03:30.567890Z 115 [Note] Query OK, 1572880 rows in set
(28.15 sec)
Jul 12 04:03:50 ufw-audit[15601]: [UFW ALLOW] IN= OUT=eth0 SRC=172.17.0.5 DST=185.125.190.44
LEN=60 TOS=0x00 PREC=0x00 TTL=64 ID=12345 DF PROTO=TCP SPT=48128 DPT=443 WINDOW=65535 RES=0x00 SYN
Jul 12 04:06:30 sshd[15482]: pam_unix(sshd:session): session closed for user db_admin
Jul 12 04:06:50 systemd[1]: Starting Clean up of User Sessions...
Jul 12 04:07:05 systemd[1]: Removed slice User Slice of UID 1001.
```

Figure 5: Log Example of Data Exfiltration (Simplified Version)

## G  COMPARED DATASETS

**Blue Gene/L Dataset.**  The Blue Gene/L dataset is provided by Lawrence Livermore National Laboratory (LLNL). The dataset comprises approximately 348,460 alert messages, spanning 41 distinct alert categories. The dataset covers a time period of 215 days, totaling approximately 1.207 GB of log data, and includes a wide range of system anomalies, from hardware failures to software errors.

**Thunderbird Dataset.**  The Thunderbird dataset is provided by Sandia National Laboratories (SNL). The dataset comprises approximately 3,248,239 alert messages, spanning 10 distinct alert categories. The dataset covers a period of 244 days, totaling approximately 27.367 GB of log data, which includes a range of system anomalies such as hardware failures, operating system issues, and network and disk system errors.

**HPC Dataset.**  The HPC dataset was collected from the Los Alamos National Laboratory, which comprises large-scale event logs from a high-performance computing cluster. Specifically, the dataset consists of 433,490 log messages and encompasses 106 distinct log formats. These logs capture various state transitions within the cluster's operations, including system configurations, environmental changes, and error messages. This dataset serves as a crucial benchmark for evaluating the performance of log analysis algorithms.

**HDFS Dataset.**  The HDFS dataset was collected from a Hadoop cluster deployed on over 200 EC2 nodes, processing more than 200 TB of random data over a 48-hour period. The dataset consists of over 24 million log entries, capturing a wide range of system operations related to distributed storage and large-scale data processing. These logs include critical information on block allocation, replication, deletion, and other key operations, providing a valuable resource for performance monitoring and anomaly detection in large-scale distributed computing environments.

**BETH Dataset.**  The BETH dataset was collected through the BPF-Extended Tracking Honeypot, containing over 8 million data records that encompass host activities and attack behaviors within a cloud environment. Specifically designed for cybersecurity anomaly detection research, the BETH dataset is primarily used for unsupervised anomaly detection tasks. The dataset includes kernel-level process invocation logs and network-level DNS query logs, with each record manually annotated to distinguish between benign and malicious behaviors. Logs for each host contain benign activity and

at most one attack event. The dataset is divided into training, validation, and test sets, making it suitable for performance evaluation of behavioral analysis and anomaly detection algorithms.

**IBM Cloud Console Dataset.** The IBM Cloud Console dataset is a large-scale telemetry dataset collected from the IBM Cloud Console over a period of 4.5 months, from January 22, 2024, to June 7, 2024. The dataset contains 39,365 rows and 117,448 columns, with each row representing a 5-minute time interval and capturing response time information from microservices across IBM Cloud data centers. Column names in the dataset represent various features such as timestamp, location, type, host ID, method, status code, and API endpoint. This dataset is intended for anomaly detection research, providing real-world cloud environment data that aids in the development and validation of anomaly detection methods applied to large-scale cloud computing systems.

**RS-Anomic Dataset.** The RS-Anomic dataset is a multivariate time-series dataset created based on the RobotShop microservice application, designed to support anomaly detection research in microservice architectures. The dataset comprises 100,464 normal instances and 14,112 anomalous instances, covering ten distinct types of anomalous behaviors, including service downtime, high concurrent user load, high CPU usage, packet loss, and response delay.

## H BASELINES

**LogBERT.** This work adopts a self-supervised Transformer that learns from normal-only log sequences via (i) masked log-key prediction (MLKP) and (ii) a hypersphere compactness loss (VHM) to encourage tight clusters of normal sequences. Raw logs are parsed into templates (e.g., Drain), then sequenced by dataset convention (*HDFS_v1*: session-based; *BGL*: 5-minute sliding windows; *ThunderBird*: 1-minute windows). Following the public setup, the authors configure a 2-layer Transformer encoder (50-d input representations, 256-d hidden states). During inference, masked keys are predicted and an anomaly is declared when observed keys fall outside the top-$g$ candidates or when the count of unexpected keys exceeds a threshold $r$. This configuration yields strong precision/recall trade-offs across all three datasets, with particularly strong results on longer sequences in *ThunderBird*.

**LogAnomaly.** The authors employ an unsupervised LSTM-based detector that jointly captures (a) *sequential* patterns (via next-template prediction) and (b) *quantitative* invariants (via template count vectors). Templates are produced by a log parser (e.g., FT-Tree) and embedded with Template2Vec to inject semantic similarity (synonyms/antonyms). The default configuration uses a window size of 20 (step 1) and two LSTM layers (128 units each), trained on the earlier portion of logs (normal-only) and evaluated on the holdout. Modeling both sequence order and template-count relations reduces false alarms and achieves high F1 on *HDFS_v1* and *BGL*; the combined view is especially beneficial when sequential cues alone are ambiguous.

**LogLLM.** This work treats windowed logs as natural-language context and applies an instruction/few-shot prompting scheme to (i) summarize state, (ii) assess deviations, and (iii) output an anomaly decision with rationale. The approach requires no dataset-specific fine-tuning; performance depends primarily on prompt design (structure, exemplars), windowing, and decoding settings (e.g., temperature). In the experiments on *HDFS_v1*, *ThunderBird*, and *BGL*, the method attains competitive recall, while precision improves when template structure, light rules, or a symbolic post-verifier are incorporated to constrain spurious triggers.

**Isolation Forest.** Isolation Forest is a machine learning algorithm used for anomaly detection, which identifies outliers by constructing a set of random trees, known as "isolation trees." The algorithm works by randomly selecting features and split values to iteratively partition the data into smaller regions. Anomalies are generally easier to isolate, leading to shorter "isolation paths." Isolation Forest is particularly effective for high-dimensional data and offers high computational efficiency.

**Decision tree.** Decision tree is a commonly used machine learning algorithm for classification and regression tasks. It makes decisions by recursively partitioning the data into different regions. At

each split, the data is divided into two subsets based on specific features and conditions, continuing until a predetermined stopping criterion is met. The structure of a decision tree resembles a tree diagram, starting from the root node and progressing through a series of splits, ultimately providing a prediction at the leaf nodes. While decision trees are easy to understand and interpret, they are prone to overfitting.

**Logistic regression.** Logistic regression is a machine learning algorithm commonly used for binary classification problems. It computes the weighted sum of input features and applies a Sigmoid function to convert the result into a probability value. This probability represents the likelihood of a particular class, typically used to determine the class to which an instance belongs. Logistic regression is simple, easy to implement, and can output probability values, which are useful for assessing the uncertainty of predictions.

**K-Means.** K-Means is a widely used unsupervised learning algorithm for clustering analysis. It partitions the data into several clusters, each represented by a centroid. The algorithm iteratively adjusts the position of the centroids to ensure that data points within each cluster are as close as possible to the centroid. This process continues until the cluster assignments no longer change. K-Means is well-suited for large-scale data and offers high computational efficiency.

**Rarity Model.** The Rarity Model is an unsupervised learning method based on data rarity, primarily used for anomaly detection. The model identifies samples that occur less frequently in the dataset and treats them as outliers. Typically, rare samples have a higher likelihood of being anomalous compared to more common samples, making the Rarity Model effective in detecting these anomalous data points.

**OOV Detector.** The OOV Detector is a method used to identify "out-of-vocabulary" (OOV) words. It works by recognizing words that were not present in the training data and marking them as OOV. The OOV Detector is commonly applied in natural language processing tasks to handle unseen vocabulary and enhance the model's ability to adapt to new words.

# I PROMPTS

```
### Anomalous Scenario: *1. CPU Hog Process*

**Description:**
One or a few processes continuously consume extremely high CPU resources due to a bug or malicious
behavior.

**Anomalous Metric Behavior:**
- **cpu usage:** `gradual increase` (sustained growth) or `spike` (sudden peak) up to 100% and
remains at that level.
- **gpu usage:** N/A
- **mem usage:** generally stable.
- **disk_io / net_in / net_out:** generally stable.

**Anomalous Log Behavior:**
- **Application logs:** request latency increases, with a large number of `timeout` errors.
- **System logs:** usually no directly related entries.

**Format Requirements:**
- CSV data must contain 90 rows (each row at 5-second intervals).
- Log data must be aligned with the CSV (timestamps must correspond).
- Log entries must use realistic formats and reflect real-world conditions.
- Include a small number of normal behavior log entries.
- Log files must **not** contain any direct descriptions of resource metrics.
- Both CSV and log files must resemble realistic operational data.
- File naming convention: `<number>.csv` and `<number>.log`.

**Note:** Two sample files are provided as references; please follow their format closely.
```

Figure 6: Prompt Example for Data Generation of CPU Hog Process (Simplified version)

```
You need to decide whether it is "anomaly" or "normal". Set the "is_anomaly" is true if the result
is anomaly, otherwise is false. Moreover, give the anomaly scenario, if it is normal choose "none".
Please only output the answer **in following json format** without reason:

```json
{
  "is_anomaly": true or false,
  "anomaly_scenario": a series of anomaly scenarios | "none"
}
```

Figure 7: Prompt Example for Vanilla LLM Prompting (Simplified version)

```
You are a metrics anomaly detection agent.

Given a time series of metric data (e.g., CPU usage, memory usage, network traffic), determine:
1. Whether the sequence contains an anomaly.
2. If so, describe the anomaly and include its **approximate time of occurrence**.
3. Identify which of the following five categories best describes the anomaly:

  [1] Spike (sudden sharp increase)
  [2] Dip (sudden sharp decrease)
  [3] Gradual Increase (slow and steady rise)
  [4] Gradual Decrease (slow and steady fall)
  [5] Fluctuation (repeated high-variance oscillation)

Respond **only** with a JSON object in the following format:

```json
{
  "is_anomaly": true or false,
  "description": "A concise description of the anomaly, including approximate time, anomaly
metrics and anomaly type in one sentence"
}
```

Figure 8: Prompt Example for Metrics Agent (Simplified version)

```
You are a log anomaly detection agent.

Given a sequence of raw system or application log entries, your task is to:
1. Analyze whether the logs indicate any abnormal or suspicious activity.
2. Summarize the **key observed behavior in a single sentence**, and include the **time or time
range** when the behavior occurred.
3. Estimate the severity of the anomaly using the following levels:
   – "high": Indicators of intrusion, privilege abuse, data exfiltration, malware activity, or
persistent service failures
   – "medium": Repeated login failures, unauthorized access, suspicious configuration changes
   – "low": Slight irregularities or uncommon but benign events

Respond **only** with a JSON object in the following format:

```json
{
  "description": "Summary of log behavior including time and anomaly type with anomaly event",
  "anomaly_level": "high" or "medium" or "low"
}
```

Figure 9: Prompt Example for Log Agent (Simplified version)

22
```

You are an **Integrated Agent** responsible for evaluating whether a cloud system case represents
a real anomaly by correlating metric and log data.

You will be given:
- The output of a **metrics agent**, in the form of:
  - `is_anomaly`: true or false
  - `description`: A short explanation including time and anomaly pattern

- The output of a **log agent**, in the form of:
  - `anomaly_level`: "low", "medium", or "high"
  - `description`: A short explanation including time and observed behavior

---

### Your Task:

1. Determine whether the system behavior represents a **true anomaly**.
2. Use **causal reasoning** to explain how the metric pattern and log behavior relate to each
other in time and implication.
3. Provide a final conclusion with the following structure:
    - A boolean `is_anomaly` value.
    - If `is_anomaly` is true, classify the event by selecting one `anomaly_category` and one
corresponding `anomaly_subtype` from the list below.
    - If `is_anomaly` is false, set both `anomaly_category` and `anomaly_subtype` to `"none"`.
    - A single-sentence `reason` that concisely combines the evidence from metrics and logs with
timestamps and your logical interpretation.

---

### Anomaly Classification: a series of anomaly scenarios

--

### Special Attention:

If the metrics agent reports an anomaly (`is_anomaly: true`) **but** the log agent reports
`"anomaly_level": "low"`:

- **Do not assume an anomaly by default.**
- First, check if the log entry provides a **reasonable explanation** for the metric behavior
(e.g., a scheduled backup, a database indexing job, or a planned deployment).
- If the log provides a benign explanation, treat the behavior as **normal** (`is_anomaly: false`).
- Only report a true anomaly if the logs fail to plausibly explain the metric spike or if they
contain subtle signs of early-stage risk despite the "low" level.

Be conservative and prioritize avoiding false positives.

---

### Output Format (JSON only):
Example for a anomaly case
```json
{
  "is_anomaly": true,
  "anomaly_category": "resource_exhaustion_and_bottleneck",
  "anomaly_subtype": "memory_leak",
  "reason": "One-sentence causal explanation combining metrics and logs results."
}
```

Example for a non-anomaly:
```json
{
  "is_anomaly": false,
  "anomaly_category": "none",
  "anomaly_subtype": "none",
  "reason": "The observed CPU spike at 14:32 UTC was caused by a scheduled daily database indexing
job, which is considered normal system behavior."
}
```

Figure 10: Prompt Example for Integrated Agent (Simplified version)

## J Symbolic Verification Example For the Mining Scenario (Simplified)

Let the last 30 time steps be $W = \{n - 29, \ldots, n\}$. Let the CPU threshold be $\tau_{\mathrm{cpu}} = 85$ and the minimum exceedance count be $k_{\min} = 24$. Let the keyword set for mining be $\mathcal{K}$ (e.g., xmrig, stratum+tcp, nanopool.org, custom-miner.service, update-run.sh, downloading payload, Accepted publickey, ufw). Let the minimum number of distinct keyword matches be $s_{\min} = 6$. Denote the multivariate metric sequence by $M$ and the multiset of log lines by $L$.

We write $k \preccurlyeq_{\mathrm{ci}} \ell$ to mean that keyword $k \in \mathcal{K}$ appears as a (case-insensitive) substring of log line $\ell \in L$.

**Metric rule (sustained high CPU).** Define

$$C_{\mathrm{cpu}}(M) = \sum_{i \in W} \mathbf{1}[M_i^{\mathrm{cpu}} > \tau_{\mathrm{cpu}}], \qquad \mathrm{VerifyMetric}_{\mathrm{mining}}(M) = \mathbf{1}[C_{\mathrm{cpu}}(M) \geq k_{\min}]. \quad (3)$$

That is, the metric-side condition passes iff at least $k_{\min} = 24$ of the last 30 CPU usage values exceed $85\%$.

**Log rule (multiple mining-related clues).** Let the set of *distinct* matched keywords be

$$\mathcal{S}(L) = \{k \in \mathcal{K} \mid \exists \ell \in L \text{ such that } k \preccurlyeq_{\mathrm{ci}} \ell\}, \qquad C_{\mathrm{log}}(L) = |\mathcal{S}(L)|. \quad (4)$$

Then the log-side condition passes iff at least $s_{\min} = 6$ distinct mining clues appear:

$$\mathrm{VerifyLog}_{\mathrm{mining}}(L) = \mathbf{1}[C_{\mathrm{log}}(L) \geq s_{\min}]. \quad (5)$$

**Final symbolic verification for mining.** The mining scenario is accepted iff both metric and log checks pass:

$$\mathrm{Verifier}_{\mathrm{mining}}(M, L) = \begin{cases} 1, & \text{if } \mathrm{VerifyMetric}_{\mathrm{mining}}(M) = 1 \ \wedge \ \mathrm{VerifyLog}_{\mathrm{mining}}(L) = 1, \\ 0, & \text{otherwise.} \end{cases} \quad (6)$$

