# OpenReview forum: "Towards Generalizable Context-aware Anomaly Detection: A Large-scale Benchmark in Cloud Environments"
_ICLR.cc/2026/Conference — Submitted to ICLR 2026_

### Official Review · Reviewer_zLDS · 2025-11-01

**Soundness:** 3
**Presentation:** 3
**Contribution:** 2
**Rating:** 4
**Confidence:** 4

**Summary:**

This paper introduces CloudAnoBench, a benchmark for context-aware anomaly detection in cloud systems that jointly contains metrics and logs across 28 anomalous and 16 deceptive normal scenarios. It also proposes CloudAnoAgent, a multi-agent LLM system with Fast/Slow Detection and a symbolic verifier that correlates metrics and logs to reduce false positives and identify scenarios. On CloudAnoBench, CloudAnoAgent improves F1 and lowers FPR versus ML baselines and vanilla LLM prompting. It also adapts to log-only datasets (HDFS v1, Thunderbird, BGL) with competitive results.

**Strengths:**

1. Positioning anomalies as contextual interactions between metrics and logs is well-motivated and distinct from point-anomaly setups. The benchmark includes deliberately deceptive normal scenarios to stress false-positive robustness.


2. The scenario taxonomy spans resource, network, software/app, malicious, and subtle cases; normal scenarios are enumerated with plausible operational events.

3. CloudAnoAgent improves F1 and reduces FPR over ML-based vanilla-LLM methods. It further shows competitive performance on log-only datasets.

**Weaknesses:**

1. The dataset generation relies heavily on LLMs. Though with human review for verification, the paper lacks quantitative evidence of how realistic the generated metric dynamics and log semantics are.

2. There is a lack of more recent DL-based and LLM-based anomaly detection baselines that are metric-only. The paper only evaluates ML-based metric-only methods.

3. The paper provides ablation with/without the symbolic verifier but does not isolate the contributions of Fast vs Slow stages or the Integrated Agent, leaving unclear where most of the gains arise.

**Questions:**

1. Dataset creation is heavily dependent on LLMs generating both metrics and logs with “code execution,” followed by GPT-4o verification and manual review. Are the metrics and logs generated from scenarios created by the LLMs but running on real hardware setups? If it is running on real hardware, how to manage the software environments of different anomaly scenarios?

2. In addition, how is each anomaly scenario in the 5 categories being selected into the benchmark? How do we ensure that they reflect the realistic cloud anomalies or incidents? How much proportion of the LLM-generated examples are inspected by human experts? Is the system based on any real-world incident traces?

3. The paper mentions that LO2 is a dataset with logs and metrics from a microservice system. How is it different than CloudAnoBench in terms of data collection, workload realism, etc?

4. Can you provide a comparison with state-of-the-art DL-based and LLM-based metrics-only methods? Current ML-based methods seem to be too simple.

5. Can you further provide an ablation study comparing the system without slow detection and without integrated agent? Currently, only one ablation with a symbolic verifier is provided.

6. What is being learned and refined in the symbolic verifier for the refinement process described in Section 4.2? Are they rules used in the metric verifier and log verifier?

7. How to deal with the context window limit for the metric and log agent if the input data is long?

---

> ### Author Response · Authors · 2025-11-21
>
> > **Weaknesses**
> > W1: The dataset generation relies heavily on LLMs. Though with human review for verification, the paper lacks quantitative evidence of how realistic the generated metric dynamics and log semantics are.
>
> Thank you for the constructive feedback. In response, we conducted an additional alignment study comparing CloudAnoBench with the real-world RS-Anomic[1] dataset. By matching CPU and network anomaly scenarios across the two datasets and evaluating their temporal traces with Spearman’s ρ, Kendall’s τ, and DTW/len, we quantified how closely CloudAnoBench reflects real anomaly behaviors. The results show consistently strong alignment across all three metrics, indicating that CloudAnoBench captures realistic temporal dynamics rather than synthetic artifacts. This analysis provides further evidence that our benchmark reliably mirrors real operational conditions.
>
> ### Experiment for Measuring Alignment Between CloudAnoBench and Real-World Distribution
>
> | Category  | Scenario                    | Spearman ρ | Kendall τ | DTW/len |
> |----------|------------------------------|------------|-----------|---------|
> | **CPU**  | High CPU Usage / CPU Hog     | **0.816**  | **0.656** | **0.179** |
> | **Network** | High Latency              | **0.688**  | **0.513** | **0.169** |
> | **Network** | Out-of-order Packets      | **0.659**  | **0.496** | **0.160** |
> | **Network** | Packet Loss               | **0.659**  | **0.494** | **0.182** |
>
> **Quantitative Study**: To evaluate whether CloudAnoBench exhibits realistic temporal behaviors rather than synthetic artifacts, we selected two anomaly categories (CPU and networking) from CloudAnoBench and matched them with their closest counterparts in the real-world RS-Anomic dataset. We then applied three sequence-similarity metrics—Spearman’s ρ, Kendall’s τ, and DTW/len—to compare the corresponding metric traces. These metrics capture complementary aspects of temporal alignment: Spearman’s ρ assesses whether two sequences share a similar monotonic trend, Kendall’s τ measures the consistency of their local ordering, and DTW/len quantifies how much non-linear temporal warping is required to align their shapes.
>
> Across both CPU and multiple network anomaly types, we observe consistently high similarity scores.
> - CPU behavior shows ρ = 0.816, τ = 0.656, and DTW/len = 0.179, indicating strong agreement in global trend, rank ordering, and overall temporal form.
> - Network anomalies similarly achieve ρ ≈ 0.66–0.69, τ ≈ 0.49–0.51, and DTW/len ≈ 0.16–0.18, suggesting that the temporal dynamics of CloudAnoBench closely follow patterns observed in operational cloud systems.
>
> **Qualitative analysis** reinforces these findings. For the same anomaly type, CloudAnoBench and RS-Anomic traces exhibit comparable temporal trajectories: CPU-hog scenarios show rapid increases followed by sustained high utilization, while network anomalies display characteristic latency buildup, jitter, and burst behavior. In all matched cases, the distributional patterns, rise-and-fall structure, and noise characteristics remain aligned across datasets. For logs, however, direct comparison is not feasible, since environment-specific details such as error identifiers differ substantially across systems.
>
> We thank the reviewer for raising this point. Due to space limitations, we were unable to include the alignment study in the main paper. As shown above, our quantitative and qualitative analyses demonstrate that CloudAnoBench closely reflects real-world metric behavior. Because logs contain environment-specific identifiers, log-level similarity is not meaningful across datasets, so we focus on the metric dynamics as an environment-agnostic basis for realism. Overall, the results confirm that CloudAnoBench preserves realistic system dynamics and aligns well with real operational data.
>
>
> [1] Akmeemana, Lahiru, et al. "GAL-MAD: Towards Explainable Anomaly Detection in Microservice Applications Using Graph Attention Networks." arXiv preprint arXiv:2504.00058 (2025).

---

> ### Author Response · Authors · 2025-11-21
>
> > W2: There is a lack of more recent DL-based and LLM-based anomaly detection baselines that are metric-only. The paper only evaluates ML-based metric-only methods.
>
> We thank the reviewer for the suggestion. Regarding more recent DL-based anomaly detection methods, we note that many of them are **highly dataset-specific** and are not designed to handle the **long-context, multimodal input** present in CloudAnoBench. Furthermore, they typically require **retraining or fine-tuning**, often with architecture changes tailored to the target dataset. As a result, these approaches cannot be directly applied to CloudAnoBench without significant re-engineering, which would make the comparison unfair and outside the scope of evaluation-focused benchmarks.
>
> In response, we conducted additional experiments, including evaluating the **metrics-only agent**, the **logs-only agent**, and ML baselines augmented with a **log parser**, allowing them to jointly process metrics and logs. All these results are provided in our responses to Q4 and Q5. Please refer to weakness 3 for the ablation experiments.

---

> ### Author Response · Authors · 2025-11-21
>
> > W3: The paper provides ablation with/without the symbolic verifier but does not isolate the contributions of Fast vs Slow stages or the Integrated Agent, leaving unclear where most of the gains arise.
>
> We have included ablation experiments evaluating the **metrics-only agent** and the **logs-only agent**. It is important to clarify that, in the main paper, the log agent was designed to output a *scenario-level likelihood distribution* rather than a binary anomaly prediction. For the purpose of this ablation study, however, both the metrics-only and logs-only variants were configured as **binary anomaly detectors** to ensure a fair, controlled comparison.
>
> In addition, we extended the evaluation by integrating a **log parser** into all ML baselines, enabling them to jointly use metrics and logs. All these results, along with the metrics-only and logs-only ablations, are provided in our responses to Q4 and Q5.

---

> ### Author Response · Authors · 2025-11-21
>
> > **Questions**
> >Q1. Dataset creation is heavily dependent on LLMs generating both metrics and logs with “code execution,” followed by GPT-4o verification and manual review. Are the metrics and logs generated from scenarios created by the LLMs but running on real hardware setups? If it is running on real hardware, how to manage the software environments of different anomaly scenarios?
>
> We thank the reviewer for raising the dataset realism concern. CloudAnoBench is generated through a combination of LLM-driven scenario construction and code-tool assisted metrics generation, and it is not executed on real hardware. However, as shown in our responses to Weakness 1 and in the public comment, our quantitative alignment study demonstrates that CloudAnoBench’s metric patterns closely match those observed in real operational data (RS-Anomic). This provides strong evidence that CloudAnoBench exhibits realistic system dynamics despite being generated synthetically.
>
> It is also important to note that **heterogeneous software environments** are common across existing cloud anomaly detection datasets[1][2][3]. As a result, datasets naturally differ in both log semantics and metric composition. For example, microservice-oriented datasets often rely on HTTP status codes and request-level indicators to reflect anomalies, whereas resource-centric datasets represent anomalies primarily through CPU, GPU, or memory usage patterns. This diversity in data sources, environments, and semantics is inherent to the domain, and CloudAnoBench follows this broader landscape by focusing specifically on **context anomalies that jointly require metrics and logs**.
>
> [1] Bakhtin, Alexander, et al. "LO2: Microservice API Anomaly Dataset of Logs and Metrics." Proceedings of the 21st International Conference on Predictive Models and Data Analytics in Software Engineering. 2025.
>
> [2] Islam, Mohammad Saiful, et al. "Anomaly detection in large-scale cloud systems: An industry case and dataset." 2025 IEEE/ACM 47th International Conference on Software Engineering: Software Engineering in Practice (ICSE-SEIP). IEEE, 2025.
>
> [3] Li, Zeyan, et al. "Constructing large-scale real-world benchmark datasets for aiops." arXiv preprint arXiv:2208.03938 (2022).

---

> ### Author Response · Authors · 2025-11-21
>
> > Q2. How is each anomaly scenario in the five categories selected into the benchmark? How do we ensure that they reflect realistic cloud anomalies or incidents? How much proportion of the LLM-generated examples are inspected by human experts? Is the system based on any real-world incident traces?
>
> To construct the anomaly scenarios in CloudAnoBench, we first collected and reviewed a wide range of scenarios reported in prior work[1][2]. After an initial round of manual filtering, consolidation, and deduplication, we obtained a unified set of anomaly scenarios. Human experts then categorized each scenario into the appropriate anomaly category.
>
> Because many real-world anomaly scenarios cannot be safely reproduced due to reliability risks, service-impact concerns, and operational cost constraints, it is infeasible to generate such data directly on production systems. To address this limitation, we adopt a more practical and scalable approach by using LLMs to assist with dataset generation. This enables us to systematically cover a broad spectrum of anomaly types while maintaining consistency and controllability in the scenario design.
>
> [1] Peter Garraghan, Paul Townend, and Jie Xu. An empirical failure-analysis of a large-scale cloud computing environment. In 2014 IEEE 15th International Symposium on High-Assurance Systems Engineering, pp. 113–120, 2014. doi: 10.1109/HASE.2014.24.
>
> [2] Kate Highnam, Kai Arulkumaran, Zachary Hanif, and Nicholas R Jennings. Beth dataset: Real cybersecurity data for unsupervised anomaly detection research. In CEUR Workshop Proc, volume 3095, pp. 1–12, 2021.

---

> ### Author Response · Authors · 2025-11-21
>
> > Q3. The paper mentions that LO2 is a dataset with logs and metrics from a microservice system. How is it different from CloudAnoBench in terms of data collection, workload realism, etc.?
>
> CloudAnoBench is constructed through an LLM-assisted, code-driven generation process, followed by systematic verification by human experts. In contrast, the LO2 dataset is designed specifically for anomaly detection in microservice systems. LO2 is built on an open-source, production-ready microservice platform (Light-OAuth2), where Locust is used to generate API traffic that includes both normal requests and faulty ones (e.g., incorrect parameters or malformed calls).
>
> A key distinction lies in the type and generality of the metric data. CloudAnoBench focuses on **resource-oriented metrics** that are broadly representative across a wide variety of cloud environments, such as CPU usage, GPU usage, memory pressure, and similar system-level indicators. Its logs capture diverse server-side operational events and system responses. LO2, by contrast, derives its “metric” signals by aggregating HTTP status code counts over time windows, and its logs are standardized into labeled templates that reflect microservice-specific request handling behavior.
>
> These structural differences reflect the different goals of the two datasets:
> LO2 targets microservice API-level failure patterns, while CloudAnoBench aims to model **context anomalies** that require joint reasoning over general-purpose system metrics and logs across heterogeneous cloud scenarios.
>
> Bakhtin, Alexander, et al. "LO2: Microservice API Anomaly Dataset of Logs and Metrics." Proceedings of the 21st International Conference on Predictive Models and Data Analytics in Software Engineering. 2025.

---

> ### Author Response · Authors · 2025-11-21
>
> > Q4. Can you provide a comparison with state-of-the-art DL-based and LLM-based metrics-only methods? Current ML-based methods seem to be too simple.
> > Q5. Can you further provide an ablation study comparing the system without slow detection and without integrated agent? Currently, only one ablation with a symbolic verifier is provided.
>
> Thank you for the question. In our work, we observed that many recent state-of-the-art deep learning–based anomaly detection methods are highly dataset-specific and can only operate effectively on a narrow set of datasets. In contrast, CloudAnoBench contains cases where both metrics and logs must be processed jointly, and the log sequences can be substantially longer than what most of these models are designed to handle.
>
> As a result, these methods either cannot run directly on CloudAnoBench, or they require substantial retraining, architectural modification, and hyperparameter tuning, which would make the comparison unfair and methodologically inconsistent. To address the reviewer’s concern and ensure fair evaluation within the scope of our benchmark, we added:
>
> - ML algorithms with log parser performance, enabling ML baselines to use both metrics and logs.
> - Ablation studies for metrics-only and logs-only agents, which further clarify the contribution of each modality.
>
> These additions allow us to provide meaningful and comparable baselines under the constraints of CloudAnoBench’s multimodal and long-context design.
>
> ### ML Algorithms with log parser performance
>
> | Model                     | P     | R     | FPR   | F1    | Acc   |
> |---------------------------|-------|-------|-------|-------|--------|
> | IsolationForest           | 54.0  | 98.1  | 100.0 | 69.7  | 53.5  |
> | KMeans                    | 55.3  | 93.0  | 90.0  | 69.4  | 55.2  |
> | RarityModel               | 76.5  | 32.8  | 12.1  | 45.9  | 57.9  |
> | OOV Detector              | 60.4  | 63.2  | 49.6  | 61.8  | 57.4  |
> | CloudAnoAgent (w/o verifier)  | 86.8  | 92.0  | 16.9  | 89.3  | 73.2  |
> | CloudAnoAgent (with verifier) | 89.6  | 93.1  | 12.9  | 91.3  | 76.3  |
>
>
> ### Ablation study for metrics agent only and log agent only
>
> **CloudAnoAgent (Metrics Agent)**
> | Model                | P     | R     | FPR   | F1    | Acc   |
> |---------------------|-------|-------|-------|-------|--------|
> | Gemini 2.5 Flash     | 58.55 | 93.27 | 79.26 | 71.94 | 0.426 |
> | Gemini 2.5 Flash Lite| 59.31 | 93.70 | 77.15 | 72.64 | 0.407 |
> | Gemini 2.0 Flash Lite| 57.57 | 87.99 | 77.86 | 69.60 | 0.353 |
> | GPT-4o Mini          | 57.14 | 86.68 | 78.03 | 68.88 | 0.387 |
> | GPT-4o               | 59.04 | 89.90 | 74.87 | 71.27 | 0.411 |
> | Qwen Plus            | 59.10 | 90.34 | 75.04 | 71.45 | 0.382 |
>
> **CloudAnoAgent (Log Agent)**
> | Model                | P     | R     | FPR   | F1    | Acc   |
> |---------------------|-------|-------|-------|-------|--------|
> | Gemini 2.5 Flash     | 76.28 | 85.21 | 31.81 | 80.50 | 0.543 |
> | Gemini 2.5 Flash Lite| 76.93 | 84.48 | 30.40 | 80.53 | 0.518 |
> | Gemini 2.0 Flash Lite| 78.55 | 80.97 | 26.54 | 79.74 | 0.452 |
> | GPT-4o Mini          | 78.68 | 80.53 | 26.19 | 79.59 | 0.539 |
> | GPT-4o               | 78.17 | 83.89 | 28.12 | 80.93 | 0.574 |
> | Qwen Plus            | 78.13 | 82.14 | 27.59 | 80.09 | 0.553 |
>
> To ensure a fair evaluation, we conducted additional experiments including the **metrics-only agent**, the **logs-only agent**, and ML baselines augmented with a **log parser**, enabling them to jointly process metrics and logs. The full results are provided in the tables above.
>
> These experiments show that while log-augmented ML baselines benefit from additional information, they still suffer from high false positive rates and limited F1 scores. Meanwhile, the metrics-only and logs-only variants of CloudAnoAgent demonstrate the complementary importance of both modalities, further validating the design of CloudAnoBench and CloudAnoAgent for context-aware anomaly detection.
>
> Regarding more recent deep-learning-based anomaly detection methods, most of them are **highly dataset-specific**. They typically rely on short, fixed-length inputs (e.g., request traces, short log windows), making them incompatible with the **long-context, multi-modal input structure** of CloudAnoBench. Moreover, these methods generally require **retraining, architecture adaptation, and hyperparameter tuning** tailored to their target dataset. Applying them directly to CloudAnoBench would require substantial re-engineering and would not constitute a fair comparison within the scope of our benchmark.

---

> ### Author Response · Authors · 2025-11-21
>
> > Q6. What is being learned and refined in the symbolic verifier for the refinement process described in Section 4.2? Are they rules used in the metric verifier and log verifier?
>
> Thank you for the question — we agree that the description in the paper could have been clearer. The rules used in the symbolic verifier are **static rather than learned**. Both the metric rules and log rules were first generated by an LLM and subsequently reviewed and refined by human experts. Importantly, the experts **did not have access to the dataset distribution** during the review process, ensuring that the resulting rules do not overfit CloudAnoBench.
>
> These finalized rules are then applied within both components of the symbolic verifier — the **metrics checker** and the **logs checker** — where they serve as high-level consistency constraints to reduce major misclassifications without encoding dataset-specific patterns.

---

> ### Author Response · Authors · 2025-11-21
>
> >Q7. How to deal with the context window limit for the metric and log agent if the input data is long?
>
> Thank you for raising this important question. In our current work, the context length required for CloudAnoAgent falls well within the processing limits of modern LLMs. Looking ahead, although LLMs continue to support increasingly larger context windows, this also comes with rising computational cost and latency.
>
> Recent studies have shown that LLMs often struggle when relevant information is buried inside long documents, leading to missed or incorrect reasoning[1]. A common practice in real production systems is therefore to reduce the effective context window through various chunking strategies, which limit the amount of text the model must process at once. Widely used approaches include fixed-size chunking, document-structure–aware chunking, and contextual chunking. The latter, Contextual Chunking with LLMs, is based on the “Contextual Retrieval” method proposed by Anthropic[2], which allows the model to segment long inputs into semantically coherent units before inference.
>
> In real deployment scenarios where cost and responsiveness must be controlled, a common strategy—also used broadly in other deep learning methods—is to **chunk long sequences into smaller segments**, perform multiple inference passes, and then aggregate the results (e.g. documents[3]). This approach scales naturally to longer contexts while keeping memory usage and latency manageable. CloudAnoAgent can adopt the same strategy when deployed in environments with significantly longer metric–log histories.
>
> [1] Liu, Nelson F., et al. "Lost in the middle: How language models use long contexts." Transactions of the Association for Computational Linguistics 12 (2024): 157-173.
>
> [2] https://www.anthropic.com/engineering/contextual-retrieval
>
> [3] https://developer.nvidia.com/blog/finding-the-best-chunking-strategy-for-accurate-ai-responses/

---

### Official Review · Reviewer_PFCn · 2025-11-02

**Soundness:** 3
**Presentation:** 3
**Contribution:** 3
**Rating:** 6
**Confidence:** 3

**Summary:**

The paper introduces CLOUDANOBENCH, a large-scale benchmark for context-aware anomaly detection in cloud environments. It emphasizes realistic, multimodal context anomalies and challenging normal cases that often cause false positives. The authors propose CLOUDANOAGENT, an LLM-based multi-agent system with symbolic verification, which integrates metric and log data for robust anomaly detection and scenario identification. Experiments show CLOUDANOAGENT outperforms traditional machine learning, log-only, and vanilla LLM baselines.

**Strengths:**

+) A comprehensive and realistic benchmark that combines both metrics and logs

+) Extensive experimental results showing significant improvements over LLM baselines

**Weaknesses:**

-) The benchmark, while large, is still limited to the scenarios and data sources curated by the authors, which may not be fully covering the network anomalies

-) It might be better to analyze failure cases

**Questions:**

a) The symbolic verification part seems remedy of the agent system. Is it possible to make the agent detector stronger and eliminate symbolic verification in the future versions?

b) What is the overhead of the proposed anomaly detection system?

---

> ### Author Response · Authors · 2025-11-21
>
> > **Weaknesses**
> > W1: The benchmark, while large, is still limited to the scenarios and data sources curated by the authors, which may not be fully covering the network anomalies.
>
> We thank the reviewer for this valuable suggestion. We agree that CloudAnoBench does not cover every possible scenario within the broad space of network anomalies. In this work, our focus is on addressing gaps that existing cloud anomaly datasets do not cover—specifically:
> 1. providing a benchmark that jointly includes metrics and logs for **context anomalies**, rather than point anomalies alone;
> 2. offering **fine-grained scenario labels** beyond coarse anomaly categories;
> 3. incorporating **deceptive normal cases**, which enable meaningful evaluation of false positive robustness, a critical but underexplored dimension in previous datasets.
>
> Therefore, achieving full coverage of all anomaly scenarios across all categories was not the primary goal of this release. We appreciate the reviewer’s suggestion and will incorporate additional scenarios in future extensions of CloudAnoBench.

---

> ### Author Response · Authors · 2025-11-21
>
> >W2: It might be better to analyze failure cases.
>
> Due to space constraints, we were unable to include our failure-case analysis in the main paper. However, the symbolic verifier was introduced **precisely as a result** of analyzing CloudAnoAgent’s failure modes. Although the fast–slow detection mechanism significantly improves robustness and scenario identification, we observed that LLMs still exhibit inherent randomness, and some benchmark cases are intentionally highly deceptive. These factors occasionally lead to misclassification of scenarios or even the broader anomaly category. To mitigate these errors, we designed the symbolic verifier, which enforces general rule-based constraints and reduces misclassification at the category level. We will include a more detailed discussion of these failure cases in the next iteration of the paper.

---

> ### Author Response · Authors · 2025-11-21
>
> > **Questions**
> > a) The symbolic verification part seems remedy of the agent system. Is it possible to make the agent detector stronger and eliminate symbolic verification in the future versions?
>
> The symbolic verifier is designed to enforce a small set of general, scenario-level rules. These rules are **static rather than learned**, and both the metric rules and log rules were first generated by an LLM and then reviewed by human experts. Crucially, the reviewers did **not** have access to the dataset distribution during this process, ensuring that the rules do not overfit CloudAnoBench in any way.
>
> Because LLM-based agent systems inherently exhibit stochastic output behavior, occasional inconsistencies can arise even with strong reasoning capabilities. As the agent system continues to improve, it is indeed possible that the symbolic verifier may gradually become unnecessary. Nevertheless, the symbolic verifier’s rule set is inherently scalable and lightweight, meaning it does not introduce deployment challenges in real-world environments. Instead, it provides a stability layer that can be removed or minimized as the underlying agent model becomes more robust.

---

> ### Author Response · Authors · 2025-11-21
>
> >b) What is the overhead of the proposed anomaly detection system?
>
> ### **Latency Performance on CloudAnoBench**
>
> **ML Algorithms**
> | Model           | Mean ± Std Dev    |
> |----------------|-------------------|
> | IsolationForest| 0.61 ms ± 0.59 ms |
> | KMeans         | 0.25 ms ± 0.46 ms |
> | RarityModel    | 1.59 ms ± 0.78 ms |
> | OOV Detector   | 1.15 ms ± 0.68 ms |
>
> **Vanilla Prompting**
> | Model               | Total Latency |
> |---------------------|---------------|
> | Gemini 2.5 Flash     | 7.514 s       |
> | Gemini 2.5 Flash Lite| 0.577 s       |
> | Gemini 2.0 Flash     | 0.671 s       |
> | GPT-4o               | 2.917 s       |
> | GPT-4o Mini          | 1.834 s       |
> | Qwen Plus            | 2.761 s       |
>
> **CloudAnoAgent**
> | Model               | Metrics Agent | Log Agent | Integrated Agent | Symbolic Verifier | Total  |
> |---------------------|---------------|-----------|------------------|--------------------|--------|
> | Gemini 2.5 Flash     | 3.78 s        | 2.91 s    | 3.45 s           | 0.42 s             | 10.56 s|
> | Gemini 2.5 Flash Lite| 0.63 s        | 0.59 s    | 0.71 s           | 0.42 s             | 2.35 s |
> | Gemini 2.0 Flash     | 1.08 s        | 0.82 s    | 1.05 s           | 0.39 s             | 3.34 s |
> | GPT-4o               | 2.33 s        | 1.92 s    | 2.09 s           | 0.41 s             | 6.75 s |
> | GPT-4o Mini          | 1.35 s        | 1.09 s    | 1.23 s           | 0.40 s             | 4.07 s |
> | Qwen Plus            | 5.67 s        | 2.75 s    | 5.29 s           | 0.42 s             | 14.14 s|
>
> Despite using LLMs as the foundation of CloudAnoAgent, we acknowledge that its detection latency is higher than that of traditional ML baselines. As shown in the latency tables above, this difference reflects a fundamental **performance–latency tradeoff**. While ML baselines achieve very low latency, they also suffer from extremely high false positive rates. In real operational settings, such high FPR leads to substantial downstream costs, including wasted engineering time, unnecessary escalations, and increased resource overhead.
>
> In contrast, CloudAnoAgent delivers **substantially more accurate anomaly detection and precise scenario identification**, enabling engineers to localize root causes more efficiently despite the additional inference time. This results in lower overall operational cost and faster mean-time-to-resolution in practice.
>
> We also note that, in our experiments, we intentionally used relatively large LLMs through **remote API endpoints**, which naturally increases latency. With smaller locally deployed LLMs, model distillation, quantization, or caching strategies, the end-to-end latency can be significantly reduced. We view these engineering optimizations as promising directions for future work.

---

### Official Review · Reviewer_mBi2 · 2025-11-02

**Soundness:** 3
**Presentation:** 3
**Contribution:** 2
**Rating:** 4
**Confidence:** 3

**Summary:**

The paper introduces two main contributions for context-aware anomaly detection in cloud environments. The first is CLOUDANOBENCH, a new large-scale benchmark dataset that combines both metric and log data. A key feature of this benchmark is its inclusion of 28 anomaly scenarios and 16 "deceptive normal" scenarios, which are designed to look like anomalies but are benign, making the detection task more challenging. The second contribution is CLOUDANOAGENT, an LLM-based, multi-agent system designed to work with this kind of multi-modal data. The agent uses a "Fast and Slow" detection mechanism to process metrics and logs, and critically, it incorporates a symbolic verifier to validate the LLM's findings, aiming to improve accuracy and reduce false positives. The experimental results show that CLOUDANOAGENT outperforms traditional machine learning methods and standard LLM prompting on the new benchmark, particularly in reducing the false positive rate.

**Strengths:**

1. The benchmark itself is a solid contribution. The focus on "context anomalies" that require both metrics and logs is well-motivated, and the inclusion of "deceptive normal scenarios" addresses a common and practical challenge in real-world operations where false alarms are costly.
2. The design of the CLOUDANOAGENT, which combines LLM-based reasoning with a symbolic verifier, is a strong point. This hybrid approach directly tackles the reliability issues often associated with LLMs, using a rule-based system as a check. The significant drop in the FPR shown in Table 2 for the agent with the verifier supports the value of this design.
3. The paper provides a very clear explanation of the problem and its proposed solution. The data generation process, agent architecture, and experimental setup are well-documented. The public release of the benchmark is also a valuable service to the community.

**Weaknesses:**

1. The benchmark is generated synthetically using LLMs. While this is a practical necessity for scenarios that are dangerous to reproduce, it raises questions about the data's realism and potential biases. An LLM-based evaluation model (CLOUDANOAGENT) might perform very well on data generated by another LLM, but it's unclear if this performance would hold on real-world, non-synthetic data.
2. The claim of "strong generalization" seems a bit of a stretch. The agent is tested on older, log-only datasets (BGL, Thunderbird, HDFS_v1) in Table 4. While it performs competitively, these datasets are for point-anomaly detection from a single modality. This doesn't fully validate the agent's core strength, which is meant to be context-aware detection on multi-modal data. Generalization should ideally be shown on a different context-aware dataset.
3. The practical aspects of the CLOUDANOAGENT system, such as latency and computational cost, are not discussed. The multi-step process involving multiple agents and a verifier (Figure 3) seems like it could be slow, which is a critical factor for a real-time anomaly detection system.

**Questions:**

1. Regarding the dataset generation: The use of an LLM to generate the benchmark data is clever, but I'm curious about the potential for model-specific artifacts. Could the authors comment on whether the CLOUDANOAGENT (which also uses LLMs) might have an inherent advantage on this dataset compared to non-LLM methods? Were any steps taken, beyond the manual review mentioned in Section 3.3, to ensure the data distribution truly mirrors real-world operational data?
2. Regarding the evaluation of baselines: In Section 5.3 (Line 385), it's stated that the ML methods were restricted to using only metric data. Given that the benchmark's main premise is the combination of metrics and logs, this seems to put those baselines at a significant disadvantage. Was there any attempt to incorporate log features for the ML models, for instance through log parsing and feature extraction, to allow for a more direct comparison?
3. Regarding the symbolic verifier: This component appears to be key for the performance improvement, especially in reducing false positives. Could the authors elaborate on the process for creating the verification rules for each of the 28 anomaly scenarios? Does this require significant manual effort and domain expertise for each new anomaly type the system needs to detect? How might this approach scale in a real-world environment where new and unexpected failure modes can emerge?
4. Regarding latency: The agent's architecture involves several sequential steps. Could the authors provide some information on the end-to-end latency of the detection process, from receiving the data to producing a final decision? How does this potentially compare to the much simpler vanilla LLM or ML baselines, and what are the trade-offs for the improved accuracy?

---

> ### Author Response · Authors · 2025-11-21
>
> > **Weaknesses**
> > W1: The benchmark is generated synthetically using LLMs. While this is a practical necessity for scenarios that are dangerous to reproduce, it raises questions about the data's realism and potential biases. An LLM-based evaluation model (CLOUDANOAGENT) might perform very well on data generated by another LLM, but it's unclear if this performance would hold on real-world, non-synthetic data.
>
> Thank you for the constructive feedback. In response, we conducted an additional alignment study comparing CloudAnoBench with the real-world RS-Anomic[1] dataset. By matching CPU and network anomaly scenarios across the two datasets and evaluating their temporal traces with Spearman’s ρ, Kendall’s τ, and DTW/len, we quantified how closely CloudAnoBench reflects real anomaly behaviors. The results show consistently strong alignment across all three metrics, indicating that CloudAnoBench captures realistic temporal dynamics rather than synthetic artifacts. This analysis provides further evidence that our benchmark reliably mirrors real operational conditions.
>
> ### **Experiment: Measuring Alignment Between CloudAnoBench and Real-World Distribution**
>
> | Category | Scenario | Spearman ρ | Kendall τ | DTW/len |
> |----------|----------|-------------|------------|-----------|
> | **CPU** | High CPU Usage / CPU Hog | **0.816** | **0.656** | **0.179** |
> | **Network** | High Latency | **0.688** | **0.513** | **0.169** |
> | **Network** | Out-of-order Packets | **0.659** | **0.496** | **0.160** |
> | **Network** | Packet Loss | **0.659** | **0.494** | **0.182** |
>
> **Quantitative Analysis**: To evaluate whether CloudAnoBench exhibits realistic temporal behaviors rather than synthetic artifacts, we selected two anomaly categories (CPU and networking) from CloudAnoBench and matched them with their closest counterparts in the real-world RS-Anomic dataset. We then applied three sequence-similarity metrics—Spearman’s ρ, Kendall’s τ, and DTW/len—to compare the corresponding metric traces. These metrics capture complementary aspects of temporal alignment: Spearman’s ρ assesses whether two sequences share a similar monotonic trend, Kendall’s τ measures the consistency of their local ordering, and DTW/len quantifies how much non-linear temporal warping is needed to align their shapes.
>
> Across both CPU and multiple network anomaly types, we observe consistently high similarity scores.
> - CPU behavior shows ρ = 0.816, τ = 0.656, and DTW/len = 0.179, indicating strong agreement in global trend, rank ordering, and overall temporal form.
> - Network anomalies similarly achieve ρ ≈ 0.66–0.69, τ ≈ 0.49–0.51, and DTW/len ≈ 0.16–0.18, suggesting that the temporal dynamics of CloudAnoBench closely follow patterns observed in operational cloud systems.
>
> **Qualitative analysis** reinforces these findings. For the same anomaly type, CloudAnoBench and RS-Anomic traces exhibit comparable temporal trajectories: CPU-hog scenarios show rapid increases followed by sustained high utilization, while network anomalies display characteristic latency buildup, jitter, and burst behavior. In all matched cases, the distributional patterns, rise-and-fall structure, and noise characteristics remain aligned across datasets. For logs, however, direct comparison is not feasible, since environment-specific details such as error identifiers differ substantially between datasets.
>
> Together, these results show that although CloudAnoBench is constructed with LLM-assisted scenario design, it preserves realistic system dynamics and achieves a close match to real-server data both quantitatively and qualitatively. This strong alignment across both quantitative metrics and qualitative behaviors supports the reliability of our benchmark and strengthens the validity of CloudAnoAgent’s performance on CloudAnoBench.
>
> [1] Akmeemana, Lahiru, et al. "GAL-MAD: Towards Explainable Anomaly Detection in Microservice Applications Using Graph Attention Networks." arXiv preprint arXiv:2504.00058 (2025).

---

> ### Author Response · Authors · 2025-11-21
>
> > W2: The claim of “strong generalization” seems a bit of a stretch. The agent is tested on older, log-only datasets (BGL, Thunderbird, HDFS_v1) in Table 4. While it performs competitively, these datasets are for point-anomaly detection from a single modality. This doesn't fully validate the agent's core strength, which is meant to be context-aware detection on multi-modal data.
>
> In the domain of cloud and microservice anomaly detection, existing datasets vary widely due to the lack of a unified standard, consistent deployment environment, and common set of collected metrics or logs. Moreover, to the best of our knowledge, there is no dataset comparable to CloudAnoBench that simultaneously provides both metrics and logs and explicitly targets context anomalies. Because CloudAnoAgent is specifically designed for such multimodal, context-dependent settings, it naturally relies on domain-specific information that is absent from metrics-only, log-only, or point-anomaly datasets. As a result, it is not realistic to expect CloudAnoAgent to achieve its best performance on datasets that do not contain the contextual structure it is designed to reason over.
>
> For this reason, evaluating CloudAnoAgent solely on metrics-only or log-only datasets cannot fully demonstrate its strengths. Nonetheless, as shown in Section 6, CloudAnoAgent still achieves strong performance on BGL, Thunderbird, and HDFS_v1—datasets that contain only logs and focus on point anomalies—indicating that the model generalizes reasonably well even outside the context-aware setting it is built for. These results further support that CloudAnoAgent is not overfitted to CloudAnoBench but remains effective on datasets that were not specifically designed for it.

---

> ### Author Response · Authors · 2025-11-21
>
> > W3: The practical aspects of the CLOUDANOAGENT system, such as latency and computational cost, are not discussed. The multi-step process involving multiple agents and a verifier (Figure 3) seems like it could be slow, which is a critical factor for a real-time anomaly detection system.
>
> Thank you for raising this point. We agree that LLM-based methods generally incur higher latency compared to traditional machine learning approaches, and we explicitly acknowledge this limitation in the paper. However, given the relevance of latency in anomaly detection, we conducted an additional latency analysis across three settings: (1) ML baselines, (2) vanilla prompting, and (3) CloudAnoAgent. The detailed results are shown below.
>
> ### **Latency Performance on CloudAnoBench**
>
> **ML Algorithms**
> | Model           | Mean ± Std Dev    |
> |----------------|-------------------|
> | IsolationForest| 0.61 ms ± 0.59 ms |
> | KMeans         | 0.25 ms ± 0.46 ms |
> | RarityModel    | 1.59 ms ± 0.78 ms |
> | OOV Detector   | 1.15 ms ± 0.68 ms |
>
> **Vanilla Prompting**
> | Model               | Total Latency |
> |---------------------|---------------|
> | Gemini 2.5 Flash     | 7.514 s       |
> | Gemini 2.5 Flash Lite| 0.577 s       |
> | Gemini 2.0 Flash     | 0.671 s       |
> | GPT-4o               | 2.917 s       |
> | GPT-4o Mini          | 1.834 s       |
> | Qwen Plus            | 2.761 s       |
>
> **CloudAnoAgent**
> | Model               | Metrics Agent | Log Agent | Integrated Agent | Symbolic Verifier | Total  |
> |---------------------|---------------|-----------|------------------|--------------------|--------|
> | Gemini 2.5 Flash     | 3.78 s        | 2.91 s    | 3.45 s           | 0.42 s             | 10.56 s|
> | Gemini 2.5 Flash Lite| 0.63 s        | 0.59 s    | 0.71 s           | 0.42 s             | 2.35 s |
> | Gemini 2.0 Flash     | 1.08 s        | 0.82 s    | 1.05 s           | 0.39 s             | 3.34 s |
> | GPT-4o               | 2.33 s        | 1.92 s    | 2.09 s           | 0.41 s             | 6.75 s |
> | GPT-4o Mini          | 1.35 s        | 1.09 s    | 1.23 s           | 0.40 s             | 4.07 s |
> | Qwen Plus            | 5.67 s        | 2.75 s    | 5.29 s           | 0.42 s             | 14.14 s|
>
> As we are calling these off-the-shelf models via APIs, the latency also resulted from network latency, restricted computation resources, etc. For deployment in production, the latency should be considerably lower once using in-house deployed LLMs and with more computation resources. While CloudAnoAgent indeed has higher detection latency than traditional ML baselines, this reflects a performance–latency trade-off. ML models offer very low latency but suffer from high false positive rates, which in real deployments translate into substantial operational costs and alert fatigue. In contrast, CloudAnoAgent not only achieves significantly lower false positive rates but also identifies the underlying anomaly scenario, enabling faster and more accurate root-cause analysis. This capability is far more valuable in practical cloud and microservice environments, where misdiagnosis or delayed understanding of incidents often incurs far greater cost than a few additional seconds of detection latency.

---

> ### Author Response · Authors · 2025-11-21
>
> > **Questions**
> > Q1. **Dataset generation:** The use of an LLM to generate the benchmark data is clever, but I'm curious about the potential for model-specific artifacts. Could the authors comment on whether the CLOUDANOAGENT (which also uses LLMs) might have an inherent advantage on this dataset compared to non-LLM methods? Were any steps taken, beyond the manual review mentioned in Section 3.3, to ensure the data distribution truly mirrors real-world operational data?
>
> A1.1 We appreciate the reviewer’s concern about potential model–generator affinity bias. Prior work on synthetic data and instruction tuning [1][2] has noted that LLMs may appear to perform better when the evaluation data is produced by a similar model family. To test whether such effects exist in our setting, we evaluated a “vanilla LLM prompting” baseline that uses the same underlying LLM without any multi-agent structure or verification. If CloudAnoBench inherently favored LLM-style artifacts, this baseline should have achieved strong performance. Instead, it performs substantially worse than CloudAnoAgent (−13.7% F1), indicating that the gains do not come from shared LLM lineage but from CloudAnoAgent’s structured reasoning pipeline and symbolic verification.
>
> A1.2 We further highlight that CloudAnoBench is not directly generated by an LLM in the form of raw metric values. For the metrics, the LLM is only used to produce executable Python code based on high-level anomaly patterns (e.g., spikes, drifts, resource contention). The resulting time-series data are obtained by running this code, not by sampling from the LLM. This design explicitly decouples metric behavior from LLM textual outputs, ensuring that the dynamics are governed by programmatic simulation rather than language-model artifacts. This yields statistically grounded, operationally plausible metric traces and mitigates the risk of model-specific bias.
>
>
>
> [1] Zhou, Chunting, et al. "Lima: Less is more for alignment." Advances in Neural Information Processing Systems 36 (2023): 55006-55021.
>
> [2] Burns, Collin, et al. "Discovering latent knowledge in language models without supervision." arXiv preprint arXiv:2212.03827 (2022).

---

> ### Author Response · Authors · 2025-11-21
>
> > Q2. **Evaluation of baselines:** In Section 5.3 (Line 385), it's stated that the ML methods were restricted to using only metric data. Given that the benchmark's main premise is the combination of metrics and logs, this seems to put those baselines at a significant disadvantage. Was there any attempt to incorporate log features for the ML models, for instance through log parsing and feature extraction, to allow for a more direct comparison?
>
> ### ML Algorithms with log parser performance
>
> | Model                     | P     | R     | FPR   | F1    | Acc   |
> |---------------------------|-------|-------|-------|-------|--------|
> | IsolationForest           | 54.0  | 98.1  | 100.0 | 69.7  | 53.5  |
> | KMeans                    | 55.3  | 93.0  | 90.0  | 69.4  | 55.2  |
> | RarityModel               | 76.5  | 32.8  | 12.1  | 45.9  | 57.9  |
> | OOV Detector              | 60.4  | 63.2  | 49.6  | 61.8  | 57.4  |
> | CloudAnoAgent (w/o verifier)  | 86.8  | 92.0  | 16.9  | 89.3  | 73.2  |
> | CloudAnoAgent (with verifier) | 89.6  | 93.1  | 12.9  | 91.3  | 76.3  |
>
>
> We appreciate the reviewer’s suggestion. Log parsing is frequently incorporated into deep-learning-based anomaly detection methods, and we agree that evaluating log-augmented ML baselines provides a more complete comparison. To ensure fairness, we conducted additional experiments by integrating a standard log parser into all four ML baselines used in our study. As shown in the table above, adding the log parser leads to moderate improvements for some models. However, all ML baselines still suffer from very high false positive rates and relatively low F1 scores. This indicates that even with log parsing, traditional ML approaches remain inadequate for context-dependent anomalies, which require joint reasoning over both metrics and logs.

---

> ### Author Response · Authors · 2025-11-21
>
> > Q3. **Symbolic verifier:** This component appears to be key for the performance improvement, especially in reducing false positives. Could the authors elaborate on the process for creating the verification rules for each of the 28 anomaly scenarios? Does this require significant manual effort and domain expertise for each new anomaly type the system needs to detect? How might this approach scale in a real-world environment where new and unexpected failure modes can emerge?
>
> The symbolic verifier was introduced after analyzing numerous failure cases. While CloudAnoAgent already improves robustness through its fast–slow detection pipeline, we observed that LLM outputs still exhibit inherent randomness, and several cases in CloudAnoBench are intentionally highly deceptive. As a result, CloudAnoAgent may occasionally misclassify the anomaly scenario or even the higher-level category.
>
> The purpose of the symbolic verifier is therefore to enforce a small set of general, scenario-level rules that prevent these major misclassifications. These rules primarily help reduce false positives and incorrect category assignments; they are not designed to determine fine-grained scenario labels, nor do we expect them to do so. For this reason, the verifier uses **static, non-learned rules**. The metric rules and log rules were generated by an LLM and then reviewed and approved by human experts, even for new anomaly types, this general methodology to create new rules still holds, and therefore it is relatively easy and straightforward to build new rules. Importantly, the reviewers did **not** have access to CloudAnoBench’s data distribution during this process, ensuring that the rules do not overfit the dataset.
>
> For future work, we believe the role of a symbolic verifier is even more critical in real-world deployments. Prior research has shown that agents can autonomously extract operational rules from historical incidents, enabling a form of self-evolving rule refinement. Extending CloudAnoAgent with a similar mechanism to automatically summarize rules from past anomalies would make the verifier more adaptive and reduce dependence on manually curated rules.

---

> ### Author Response · Authors · 2025-11-21
>
> > Q4. **Latency:** The agent's architecture involves several sequential steps. Could the authors provide some information on the end-to-end latency of the detection process, from receiving the data to producing a final decision? How does this potentially compare to the much simpler vanilla LLM or ML baselines, and what are the trade-offs for the improved accuracy?
>
> As we are calling these off-the-shelf models via APIs, the latency also resulted from network latency, restricted computation resources, etc. For deployment in production, the latency should be considerably lower once using in-house deployed LLMs and with more computation resources. We appreciate the reviewer’s question and note that latency is indeed an important concern shared by many readers. In our response to Weakness 3 and in the common reply to all reviewers, we conducted additional latency experiments covering ML baselines, vanilla LLM prompting, and the full CloudAnoAgent pipeline.
>
> While CloudAnoAgent exhibits higher detection latency compared to traditional ML methods, this reflects a performance–latency tradeoff. ML baselines have very low latency but suffer from extremely high false positive rates, which translate to substantial resource overhead and human investigation cost under real operation system. In contrast, CloudAnoAgent provides far more precise anomaly detection and accurate scenario identification, enabling cloud teams to localize root causes more quickly despite the added inference time.
>
> In our experiments, we intentionally used relatively large LLMs accessed via API endpoints, which naturally increases latency. We expect latency to drop significantly with smaller locally deployed LLMs or model distillation, and we view this as a promising direction for future engineering optimization.

---

### Author Response · Authors · 2025-11-21
**To All Reviewers: Submission Updates**

First, we thank all reviewers for their valuable and constructive feedback. We also acknowledge that some answers were already included in the appendix but were not clearly signposted in the main paper. During the rebuttal phase, we have conducted additional experiments and provided more detailed clarifications for several points that were confusing or omitted due to space constraints.

In addition, we have added the following major components to the main paper and appendix to further improve the completeness and clarity of the work:

1. Latency performance results on CloudAnoBench for all evaluated methods.
2. Performance of ML algorithms with log parsers incorporated, enabling them to jointly use metrics and logs.
3. Ablation results for the metrics-only and logs-only variants of CloudAnoAgent.
4. Alignment experiments comparing CloudAnoBench against real-world data from the RS-Anomic dataset.
5. Expanded motivation and implementation details for the symbolic verifier.

All other questions are addressed individually in the detailed responses to each reviewer.

---

### Author Response · Authors · 2025-11-27

Dear Reviewer,

I hope this message finds you well! As the discussion period draws to a close with just one week left, I wanted to check in to ensure we've addressed all your concerns. If there’s anything else you’d like us to consider or discuss, please feel free to share your thoughts. We truly value your insights and want to make sure we resolve any remaining issues to make our work as strong as possible.

Thank you again for your time and effort in reviewing our paper.

---

### Meta-Review · Area_Chair_19Ks · 2026-01-11

**Summary:**

This submission introduces CloudAnoBench, a new benchmark for context anomalies in cloud environments that jointly uses metrics and logs, includes deceptive normal cases to stress false positives, and provides scenario-level labels across a reasonably broad taxonomy. Key decision-driving concerns were: (1) benchmark realism/bias due to LLM-assisted synthetic generation (risk of favoring LLM agents), (2) baseline fairness/coverage (initially weak/metric-only comparisons), (3) insufficient component attribution in the multi-agent pipeline (limited ablations), and (4) practical deployability (latency/overhead and scalability of the symbolic verifier). Despite a strong idea and a substantially improved rebuttal, the paper’s core claims about benchmark realism and agent effectiveness for multimodal context anomalies remain only partially validated, and the review scores still cluster below the acceptance threshold.

**Reviewer Concerns:**

**mBi2**
Addressed: latency/overhead now quantified; ML baselines now include log-parsing features.
Remained: synthetic benchmark realism and “generalization” claim still not fully convincing for multimodal context anomalies; verifier scalability still a question.
**zLDS**
Addressed: added metric-alignment evidence vs RS-Anomic; added log-parser ML baselines; added metrics-only / logs-only ablations.
Remained: still missing stronger modern DL baselines; still no clean isolation of fast vs slow vs integrated-agent contributions; log realism remains largely unvalidated.
**PFCn**
Addressed: partial coverage/network-scenario limitation acknowledged; overhead question answered with latency table.
Remained: failure-case analysis is still mostly narrative; verifier looks like a “patch” and long-term removal/scalability remains speculative.
**z4nP:** review appears incomplete

**Reviewer Scores:**

PFCn: 6 / confident 3.  supportive.
mBi2: 4 / confident 3. generalization concerns probably remain.
zLDS: 4 / confident 4.
Late review z4nP: appears missing

---

### Decision · Program_Chairs · 2026-01-26

Reject